# Towards Exact Gradient-based Training on Analog In-memory Computing

**Zhaoxian Wu**
Rensselaer Polytechnic Institute
Troy, NY 12180
wuz16@rpi.edu

**Tayfun Gokmen**
IBM T. J. Watson Research Center
Yorktown Heights, NY 10598
tgokmen@us.ibm.com

**Malte J. Rasch**
IBM T. J. Watson Research Center
Yorktown Heights, NY 10598
malte.rasch@googlemail.com

**Tianyi Chen**
Rensselaer Polytechnic Institute
Troy, NY 12180
chentianyi19@gmail.com

## Abstract

Given the high economic and environmental costs of using large vision or language models, analog in-memory accelerators present a promising solution for energy-efficient AI. While inference on analog accelerators has been studied recently, the training perspective is underexplored. Recent studies have shown that the "workhorse" of digital AI training - stochastic gradient descent (SGD) algorithm *converges inexactly* when applied to model training on non-ideal devices. This paper puts forth a theoretical foundation for gradient-based training on analog devices. We begin by characterizing the non-convergent issue of SGD, which is caused by the asymmetric updates on the analog devices. We then provide a lower bound of the asymptotic error to show that there is a fundamental performance limit of SGD-based analog training rather than an artifact of our analysis. To address this issue, we study a heuristic analog algorithm called Tiki-Taka that has recently exhibited superior empirical performance compared to SGD. We rigorously show its ability to *converge to a critical point exactly* and hence eliminate the asymptotic error. The simulations verify the correctness of the analyses.

## 1 Introduction

Large vision or language models have recently achieved great success in various applications. However, training large models from scratch requires prolonged durations and substantial energy consumption, which is very costly. For example, it took $2.4 million to train LLaMA [1] and $4.6 million to train GPT-3 [2]. To overcome this issue, a promising technique is application-specific hardware accelerators for neural networks, such as TPU [3], NPU [4], and NorthPole chip [5], just to name a few. Within these accelerators, the memory and processing units are physically split, which requires constant data movements between the memory and processing units. This slows down computation and limits efficiency. In this context, we focus on *analog in-memory computing (AIMC) accelerators* with resistive crossbar arrays [6, 7, 8, 9] to accelerate the ubiquitous matrix-vector multiplications (MVMs), which contribute to a significant portion of digital computation in model training and inference, e.g., about 80% in VGG16 model [10]. Very recently, the first analog AI chip has been fabricated in IBM's Albany Complex [11], and has achieved an accuracy of 92.81% on the

---

The work was supported by National Science Foundation (NSF) project 2134168, NSF CAREER project 2047177, and the IBM-Rensselaer Future of Computing Research Collaboration.

38th Conference on Neural Information Processing Systems (NeurIPS 2024).

CIFAR10 dataset while enjoying 280× and 100× efficiency and throughput value compared with the most-recent GPUs, demonstrating the transformative power of analog computing for AI [12].

In AIMC accelerators, the trainable matrix weights are represented by the conductance of the *resistive elements* in an analog crossbar array [13, 14]. Unlike standard digital devices such as GPU or TPU, trainable matrices, input, and output of MVMs in resistive crossbars are all *analog signals*, which means that they are effectively continuous physical quantities and are not quantized. In resistive crossbars, the fundamental physics (mainly Kirchhoff's and Ohm's laws) enable the devices to accelerate MVMs in both forward and backward computations. However, the analog representation of model weights requires updating weights in their unique way. To this end, an in-memory *pulse update* has been developed in [15], which changes the weights by sending consecutive pulses to implement gradient-based training algorithms like stochastic gradient descent (SGD). Pulse update reduces energy consumption and execution time significantly.

While the AIMC architecture has potential advantages, the analog signal in resistive crossbar devices is susceptible to noise and other non-ideal device characteristics [16], leading to training performance that is often suboptimal when compared to digital counterparts. Despite the increasing number of empirical studies on AIMC that aim to overcome this accuracy drop [17, 18, 19, 20], there is still a lack of theoretical studies on the performance and performance limits of SGD on AIMC accelerators.

## 1.1 Main results

The focus of this paper is fundamentally different from the vast majority of work in analog computing. We aim to build a rigorous theoretical foundation of analog training, which can uniquely characterize the performance limit of using a hardware-agnostic digital SGD algorithm for analog training and establish the convergence rate of gradient-based analog training. Consider the following standard model training problem

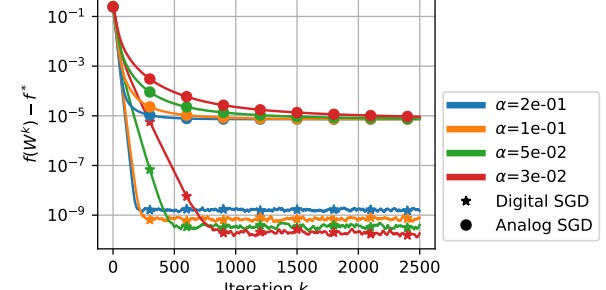

$$W^* := \arg\min_{W \in \mathbb{R}^D} f(W) \qquad (1)$$

where $f(\cdot) : \mathbb{R}^D \to \mathbb{R}$ is the objective function, $W$ is all the trainable weight stored in an analog way and $D$ is the model size.

Figure 1: Digital/Analog SGD under different learning rates.

In digital training, the workhorse algorithm for solving problem (1) is SGD, which iteratively updates weight $W$ via the following recursion

$$\text{Digital SGD} \qquad W_{k+1} = W_k - \alpha(\nabla f(W_k) + \varepsilon_k) \qquad (2)$$

where $k$ is the iteration index, $\alpha$ is a positive learning rate and $\varepsilon_k$ is the gradient noise with zero mean at iteration $k$. In digital training, the noise usually comes from the mini-batch sampling. In analog training, the noise also arises from the non-ideality of devices, including weight read noise, input/output noise, quantization noise, digital/analog conversion noise, and even thermal noise [21].

In Figure 1, we first numerically show that the asymptotic performance of SGD running on the analog devices that we term `Analog SGD` does not follow that of (2), and thus ask a natural question:

**Q1)** *How to better characterize the training trajectory of SGD on analog devices?*

Building upon the pulse update in [15, 22], in this paper, we propose the following discrete-time mathematical model to characterize the trajectory of `Analog SGD` on analog computing devices

$$\text{Analog SGD} \qquad W_{k+1} = W_k - \alpha(\nabla f(W_k) + \varepsilon_k) - \frac{\alpha}{\tau}|\nabla f(W_k) + \varepsilon_k| \odot W_k \qquad (3)$$

where $|\cdot|$ and $\odot$ represent the coordinate-wise absolute value and multiplication, respectively and $\tau$ is a device-specific parameter. We will explain the underlying rationale of this model (3) in Section 2.2. But before that, compared to (2), the extra term of (3) comes from the asymmetric update of analog devices. Following this dynamic, we prove that the `Analog SGD` only converges inexactly to a critical point, with the asymptotic error depending on the non-ideality of the analog device. Since the inexact convergence guarantee usually leads to an unfavorable result, it raises another question:

| Algorithm | Rate | Asymptotic Error |
|---|---|---|
| Digital SGD [23] | $O\left(\sqrt{\frac{\sigma^2}{K}}\right)$ | 0 |
| Analog SGD [Theorem 2] | $O\left(\sqrt{\frac{\sigma^2}{K}\frac{1}{1-W_{\max}^2/\tau^2}}\right)$ | $O(\sigma^2 S_K)$ |
| Tiki-Taka [Theorem 4] | $O\left(\sqrt{\frac{\sigma^2}{K}\frac{1}{1-33P_{\max}^2/\tau^2}}\right)$ | 0 |
| Lower bound [24] | $O\left(\sqrt{\frac{\sigma^2}{K}}\right)$ | 0 |

Table 1: Comparison between the convergence of digital and analog training algorithms: $K$ represents the number of iterations, $\sigma^2$ is the variance of stochastic gradients, $W_{\max}^2/\tau^2$ and $P_{\max}^2/\tau^2$ measure the saturation degree, and $S_K$ measures the non-ideality of analog devices (c.f. Theorem 2). Asymptotic error refers to the error that does not vanish with $K$.

**Q2)** *How to mitigate the asymptotic error induced by the asymmetric analog update?*

To answer this, we revisit a heuristic algorithm `Tiki-Taka` that has been used among experimentalists [22], and establish the *first exact convergence* result of `Tiki-Taka` on a class of AIMC accelerators.

**Our contributions.** This paper makes the following contributions (see the comparison in Table 1):

**C1)** We demonstrate that `Analog SGD` does not follow the dynamic of SGD in (2). Leveraging the underlying physics, we propose a discrete-time dynamic of the gradient-based algorithm on analog devices, and show that it better characterizes the trajectory of `Analog SGD`.

**C2)** Based on the proposed dynamic, we establish the convergence of `Analog SGD` and argue that the performance limit of `Analog SGD` is a combined effect of data noise and device asymmetry. We prove the tightness of our result by showing a matching lower bound.

**C3)** To improve the performance limit of analog training, we study a heuristic algorithm `Tiki-Taka` that serves as an alternative to `Analog SGD`. We show that `Tiki-Taka` exactly converges to the critical point by reducing the effect of asymmetric bias and noise.

**C4)** To verify the validity of our discrete-time dynamic for analog training and the tightness of our analysis, we provide simulations on both synthetic and real datasets to show that the asymptotic error of `Analog SGD` does exist, and `Tiki-Taka` outperforms `Analog SGD`.

## 1.2 Prior art

Since the seminal work on pulse update-based `Analog SGD` [15], a series of gradient-based training algorithms have been proposed to enable training on analog devices. Despite its potential energy and speed advantage, `Analog SGD` suffers from asymmetric update and noise issues, leading to large errors in training. To overcome the *asymmetric issue*, a new algorithm so-termed `Tiki-Taka` (TT-v1) introduces an auxiliary array to estimate the moving average of gradients, whose weight is then transferred to the main array periodically [22] . However, the weight transfer process between arrays still suffers from noise. To deal with this issue, TT-v2 [25] introduces an extra digital array to filter out the high-frequency noise. Enabled by these approaches, researchers successfully trained a model on the realistic prototype analog devices and reached comparable performance on a real dataset [20].

Another major hurdle comes from *noisy analog signals*, which can perturb the gradient computed by analog devices. As an alternative, a hybrid scheme that accelerates the forward and backward pass in-memory and computes gradient in the digital domain has been proposed in [17, 18], which provides a more accurate but less efficient update. In addition, the successful training by TT-v1 or TT-v2 relies on the zero-shifting technique [26], which corrects the symmetric point of devices as zero. However, the correction is inaccurate because it also involves analog signals. To deal with this issue, Chopped-TTv2 (c-TTv2) and analog gradient accumulation with dynamic reference have been proposed in [19]. Because of these efforts, analog training has empirically shown great promise in achieving a similar level of accuracy as digital training, with reduced energy consumption and training time. Despite its good performance, it is still mysterious about when and why they work.

# 2 The Physics of Analog Training

Unlike digital devices, analog devices represent the trainable weights by the conductance of the base materials, which have undergone offset and scaling due to *physical laws*. This difference leads to entirely different training dynamics on analog devices, which will be discussed in this section.

## 2.1 Revisit SGD theory and its failure in modeling analog training

This section shows that the convergence theory developed for digital SGD fails to characterize the analog training. Before that, we introduce standard assumptions for analyzing digital training.

**Assumption 1** (*L*-smoothness). *The objective $f(W)$ is L-smooth, i.e., for any $W, W' \in \mathbb{R}^D$, it holds*

$$\|\nabla f(W) - \nabla f(W')\| \leq L\|W - W'\|. \tag{4}$$

**Assumption 2** (Lower bounded). *$f(W)$ is lower bounded by $f^*$, i.e. $f(W) \geq f^*, \forall W \in \mathbb{R}^D$.*

**Assumption 3** (Noise mean and variance). *The noise $\varepsilon_k$ is independently identically distributed (i.i.d.) for $k = 0, 1, \ldots, K$, and has zero mean and bounded variance, i.e., $\mathbb{E}[\varepsilon] = 0, \mathbb{E}[\|\varepsilon\|^2] \leq \sigma^2$.*

In non-convex optimization, instead of finding a global minimum, it is usually more common to find a *critical point*, which refers to $W^*$ satisfying $\nabla f(W^*) = 0$. Under Assumptions 1–3, if $W_k$ follows the digital SGD dynamic (2), the convergence of SGD is well-understood [23, Theorem 4.8]

$$\frac{1}{K} \sum_{k=0}^{K-1} \|\nabla f(W_k)\|^2 \leq \frac{2(f(W_0) - f^*)}{\alpha K} + \alpha \sigma^2 L. \tag{5}$$

This result implies that the impact of noise can be controlled by the learning rate $\alpha$, i.e., after $K \geq \Omega(1/\alpha^2)$ iterations, the error is $O(\alpha)$. By forcing $\alpha \to 0$, the error will reduce to zero.

> **Analog SGD violates the SGD theory.** The dynamic of digital SGD exactly follows (2) up to the machine's precision. To verify whether analog training adheres to the same dynamic, we conduct a numerical simulation on a least-squares problem. We compare `Analog SGD` implemented by an analog devices simulator, IBM Analog Hardware Acceleration Kit (AIHWKIT) [27], and digital SGD implemented by PYTORCH. The same level of noise obeying Gaussian noise is injected into each algorithm. Beginning from a large learning rate $\alpha = 0.2$, we reduce the learning rate by half each time and observe the convergences. As Figure 1 illustrates, digital SGD behaves as what (5) predicts: it converges with a smaller error when a small learning rate is chosen. On the contrary, `Analog SGD` converges with a much larger error, which does not decrease as the learning rate decreases. This result demonstrates the discrepancy between the theory of digital SGD and the performance of `Analog SGD`. More details are deferred to Appendix I.

## 2.2 Training dynamic on analog devices

Compared to digital devices, the key feature of analog devices is *analog signal*. The input and output of analog arrays are analog signals, which are prone to be perturbed by noise, including read noise and input/output noise [21, 28]. Moreover, real training typically involves the utilization of mini-batch samples, which also introduces noise. Besides the data noise, another notable feature of analog accelerators is that the model weight is represented by material conductance.

**Pulse update.** To change the weights in analog devices, one needs to send an electrical *pulse* to the resistive element, and the conductance will change by a small value, which is referred to as *pulse update* [15]. To apply an update $\Delta w$ to the weight $w$, using the pulse update needs to send a series of pulses to the resistive element, the number of which is proportional to the update magnitude $|\Delta w|$. Since the increments responding to each pulse are small typically, we can regard the change in conductance as a continuous process. Consequently, common operations involving significant weight changes, like copying the weight, are expensive in analog accelerators. In contrast, gradient-based algorithms typically update small amounts at each iteration, rendering the pulse update extremely efficient. Figure 2 presents the weight change on AIMC devices with pulse number.

**Asymmetric update.** Even though the pulse update is performed efficiently inside AIMC accelerators, it suffers from a phenomenon that we refer to as *asymmetric update*. This means that if we apply

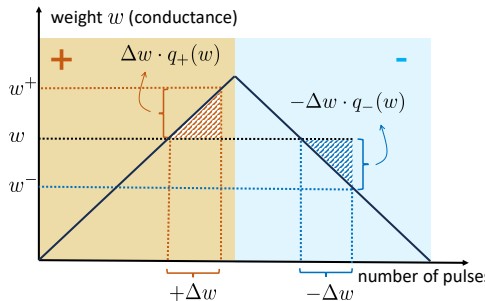 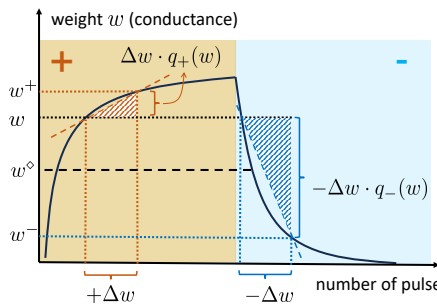

Figure 2: The weight's change with the number of pulses. Positive and negative pulses are sent continuously on the left and right half, respectively. Beginning from $w$, the weight after applying update $\Delta w$ to it is $w^+$ or $w^-$ if $\Delta w \geq 0$ or $\Delta w < 0$, respectively. The response factors $q_+(w)$ and $q_-(w)$ are approximately the slope of the curve at $w$. **(Left)** Ideal device. $q_+(w) = q_-(w) \equiv 1$. Every point is symmetric points. **(Right)** Asymmetric Linear Device (ALD). $q_+(w) = 1 - (w - w^\diamond)/\tau, q_-(w) = 1 + (w - w^\diamond)/\tau$. The symmetric point $w_\diamond$ satisfies $q_+(w^\diamond) = q_-(w^\diamond)$.

the change $\Delta w > 0$ and $\Delta w < 0$ on the same weight $w$, the amount of weight change will be different. Considering the weight $W_k$ at time $k$ and the expected update $\Delta W \in \mathbb{R}^D$, we express the asymmetric update as $W_{k+1} = U(W_k, \Delta W)$ with the update function $U : \mathbb{R} \times \mathbb{R} \to \mathbb{R}$ defined by[1]

$$U(w, \Delta w) := \begin{cases} w + \Delta w \cdot q_+(w), & \Delta w \geq 0, \\ w + \Delta w \cdot q_-(w), & \Delta w < 0, \end{cases} \tag{6}$$

where $q_+(\cdot)$ and $q_-(\cdot) : \mathbb{R} \to \mathbb{R}_+$ are up and down response factors, respectively. The *response factors* measure the ideality of analog tiles. In the ideal situation, $q_+(w) = q_-(w) \equiv 1$ (see Figure 2, left), and analog algorithms have the same numerical behavior of the digital ones. Defining the symmetric and asymmetric components as $F(w) := \frac{1}{2}(q_-(w) + q_+(w))$ and $G(w) := \frac{1}{2}(q_-(w) - q_+(w))$, the update in (6) can be expressed in a more compact form $U(w, \Delta w) = w + \Delta w \cdot F(w) - |\Delta w| \cdot G(w)$. For simplicity, assume that all the coordinates of $W$ use the same update rule, and the analog update can be written as

$$W_{k+1} = W_k + \Delta W \odot F(W_k) - |\Delta W| \odot G(W_k) \tag{7}$$

where $|\cdot|$ and $\odot$ represent the coordinate-wise absolute value and multiplication, respectively. Note that in (7), the ideal weight update $\Delta W$ is algorithm-specific: $\Delta W$ is the gradient in `Analog SGD` while it is an auxiliary weight (c.f. in Section 4) in `Tiki-Taka`.

**Remark 1** (Physical constraint). *It is attempting to scale the $\Delta w$ by $q_+(w)$ or $q_-(w)$ to cancel the effect of asymmetric update in (6) dynamically. However, this is impractical since it is hard to implement the reading and scaling at the same time on analog tiles [22].*

**Symmetric point.** The asymmetric update makes up and down responses different almost everywhere, i.e. $q_+(w) \neq q_-(w)$ for almost any $w$. If a point $w^\diamond$ satisfies $q_+(w^\diamond) = q_-(w^\diamond)$ and $G(w) = 0$, $w^\diamond$ is called a *symmetric point*. With loss of generality, the response factor is defined so that $F(w^\diamond) = 1$. Therefore, near the symmetric point, the update $\Delta w$ can be accurately applied on $w$, i.e., $U(w^\diamond, \Delta w) \approx w^\diamond + \Delta w$. If all the coordinates of matrix $W_k$ hover around the symmetric point, the analog devices can exhibit performance that resembles the digital ones. In the next section, we will show that the weight is biased toward its symmetric point.

**Asymmetric linear device.** Although our unified formulation (7) can capture the response behaviors of different materials, this paper mainly focuses on the behaviors of the asymmetric linear device (ALD), similar to the setting in [19]. ALD has a positive parameter $\tau$ which reflects the degree of asymmetry and its response factors are written as linear functions $q_+(w) = 1 - (w - w^\diamond)/\tau, q_-(w) = 1 + (w - w^\diamond)/\tau$. Consequently, ALD has $F(w) = 1$, $G(w) = (w - w^\diamond)/\tau$, and symmetric point $w^\diamond$; see Figure 2, right. Even though ALD is a simplified device model, it is representative enough to

---

[1]This paper adopts $w$ to represent the element of the weight matrix $W_k$ without specifying its index. This makes the formulations more concise and uses the fact that analog devices update all coordinates in parallel. The notation $U(W_k, \Delta W)$ on matrices $W_k$ and $\Delta W$ denote the coordinate-wise operation on $W_k$ and $\Delta W$, i.e. $[U(W_k, \Delta W)]_i := U([W_k]_i, [\Delta w]_i), \forall i \in \mathcal{I}$.

reveal the key properties and difficulties of gradient-based training algorithms on analog devices. If not otherwise specified, $w^\diamond$ is always 0 for simplicity. In summary, the update of ALD is expressed as

$$w_{k+1} = w_k + \Delta w - \frac{1}{\tau}|\Delta w| \cdot w_k \quad \text{or} \quad W_{k+1} = W_k + \Delta W - \frac{1}{\tau}|\Delta W| \odot W_k \quad (8)$$

where the first equation is the update of one coordinate while the second one stacks all the elements together. Replacing $\Delta W$ with noisy gradient $\alpha(\nabla f(W_k) + \varepsilon_k)$ reaches the dynamic (3).

## 2.3 Saturation, fast reset, and bounded weight

Based on the ALD dynamic (8), we study the properties of analog training, which serve as the stepping stone of our analyses. Recall that the asymmetric update leads to different magnitudes of increase and decrease. The following lemma characterizes the difference between two directions.

**Lemma 1** (Saturation and fast reset). *For ALD with a general $w^\diamond$, the following statements are valid*

*(Saturation)* If $\text{sign}(\Delta w) = \text{sign}(w_k)$, it holds that $|w_{k+1} - w_k| = |1 - |w_k - w^\diamond|/\tau| \cdot |\Delta w|$.

*(Fast Reset)* If $\text{sign}(\Delta w) = -\text{sign}(w_k)$, it holds that $|w_{k+1} - w_k| = |1 + |w_k - w^\diamond|/\tau| \cdot |\Delta w|$.

The proof is deferred to Section E.1. Remarkably, Lemma 1 holds for any algorithm with any update $\Delta w$, and it reveals the impact of the asymmetric update. In principle, Lemma 1 can be written as $|w_{k+1} - w_k| = |1 \pm |w_k - w^\diamond|/\tau| \cdot |\Delta w|$, where the symmetric point $w^\diamond = 0$ is omitted.

When $w_k$ is not at the symmetric point $w^\diamond = 0$, the update is scaled by a factor. When $w_k$ lies around the symmetric point, $|w_{k+1} - w_k| \approx |\Delta w|$, where all updates are applied to the weight and hence the analog devices closely mimic the performance of digital devices. When $w_k$ moves away from $w^\diamond$, i.e. $\text{sign}(\Delta w) = \text{sign}(w_k)$, the update becomes none. When $w_k$ gets closer to $\pm\tau$, nearly no update can be applied. This phenomenon is called *saturation*; see also [22]. On the contrary, $w_k$ changes faster when it moves toward $w^\diamond$, which is referred to as *fast reset*.

Because of the saturation property, the weight on the analog devices is intrinsically bounded, which will be helpful in the later analysis. The following theorem discusses the bounded property.

**Theorem 1** (Bounded weight). *Denote $\|W\|_\infty$ as the $\ell_\infty$ norm of $W$. Given $\|W_0\|_\infty \le \tau$ and for any sequence $\{\Delta W_k : k \in \mathbb{N}\}$, which satisfies $\|\Delta W_k\|_\infty \le \tau$, it holds that $\|W_k\|_\infty \le \tau, \forall k \in \mathbb{N}$.*

The proof is deferred to Section E.2. Theorem 1 claims that the weight is guaranteed to be bounded, even without explicit projection, which makes analog training different from its digital counterpart. Similar to Lemma 1, Theorem 1 does not depend on any specific analog training algorithm.

# 3 Performance Limits of Analog Stochastic Gradient Descent

After modeling the dynamic of analog training, we next discuss the convergence of `Analog SGD`. As shown by Lemma 1, the update is slow near the boundary of the active region. The weight is expected to stay within a smaller region to avoid saturation, which necessitates the following assumption.

**Assumption 4** (Bounded saturation). *There exists a positive constant $W_{\max} < \tau$ such that the weight $W_k$ is bounded in $\ell_\infty$ norm, i.e., $\|W_k\|_\infty \le W_{\max}$. The ratio $W_{\max}/\tau$ is the saturation degree.*

Assumption 4 requires that $W_k$ is bounded inside a small region, which is a mild assumption in real training. For example, one can apply a clipping operation on $w_k$ to ensure the assumption. In Appendix E.2, we show that Assumption 4 provably holds under the strongly convex assumption. It is worth pointing out that Assumption 4 implicitly assumes there are critical points in the box $\{W : \|W\|_\infty \le \tau\}$. Otherwise, the gradient will push the weight to the bound of the box and it becomes possible that $W_{\max} < \|W_k\|_\infty \le \tau$.

Intuitively, without the asymmetric bias, the weight is stable near the critical point. In contrast, with asymmetric bias, the noisy gradient that pushes $w_k$ toward its symmetric point is amplified by fast reset, while the one that drags $w_k$ away from 0 is suppressed by saturation (c.f. Lemma 1). Consequently, the weight $w_k$ is attracted by 0, which prevents the stability of `Analog SGD` around the critical point. We characterize the convergence of `Analog SGD` in the following theorem.

**Theorem 2** (Convergence of `Analog SGD`). *Under Assumption 1-4, if the learning rate is set as* $\alpha = \sqrt{\frac{f(W_0) - f^*}{\sigma^2 LK}}$ *and $K$ is sufficiently large such that $\alpha \leq \frac{1}{L}$, it holds that*

$$\frac{1}{K} \sum_{k=0}^{K-1} \mathbb{E}[\|\nabla f(W_k)\|^2] \leq O\left(\sqrt{\frac{(f(W_0) - f^*)\sigma^2 L}{K}} \frac{1}{1 - W_{\max}^2/\tau^2}\right) + \sigma^2 S_K \tag{9}$$

*where $S_K$ denotes the amplification factor given by $S_K := \frac{1}{K} \sum_{k=0}^{K} \frac{\|W_k\|_\infty^2/\tau^2}{1 - \|W_k\|_\infty^2/\tau^2} \leq \frac{W_{\max}^2/\tau^2}{1 - W_{\max}^2/\tau^2}$.*

The proof of Theorem 2 is deferred to Appendix G.1. Theorem 2 suggests that the average squared gradient norm is upper bounded by the sum of two terms: the first term vanishes at a rate of $O(\sqrt{\sigma^2/K})$ which also appears in the SGD's convergence bound (5); the second term contributes to the *asymptotic error* of `Analog SGD`, which does not vanish with the total number of iterations $K$; that is, $\limsup_{K \to \infty} \frac{1}{K} \sum_{k=0}^{K-1} \mathbb{E}[\|\nabla f(W_k)\|^2] \leq \sigma^2 S_\infty$ exist.

> **Impact of saturation/asymmetric update.** The saturation degree $W_{\max}/\tau$ affects both convergence rate and asymptotic error. The ratio is small if $W_k$ remains close to the symmetric point or $\tau$ is sufficiently large. The exact expression of $S_K$ depends on the specific noise distribution, and thus is difficult to reach. However, $S_K$ reflects the saturation degree near the critical point $W^*$ when $W_k$ converges to a neighborhood of $W^*$. Intuitively, $W_k \approx W^*$ implies $S_K \approx \frac{\|W^*\|_\infty^2/\tau^2}{1 - \|W^*\|_\infty^2/\tau^2}$. Therefore, if the critical point is near the symmetric point, the asymptotic error $S_K$ could be small. The asymmetric update has a negative impact on both rate and error. It slows down the convergence of SGD by a factor $1/(1 - W_{\max}^2/\tau^2)$, and, meanwhile, a smaller $\tau$ increases the asymptotic error.

To demonstrate the asymptotic error in Theorem 2 is not artificial, we provide a lower bound next.

**Theorem 3** (Lower bound of the error of `Analog SGD`). *There is an instance which satisfies Assumption 1-4 such that `Analog SGD` generates a sequence $\{W_k : k = 0, 1, \cdots, K\}$ which satisfies*

$$\frac{1}{K} \sum_{k=0}^{K-1} \mathbb{E}[\|\nabla f(W_k)\|^2] = \sigma^2 S_K + \Theta(\alpha) \overset{\alpha = \Theta\left(\frac{1}{\sqrt{K}}\right)}{=} \Omega\left(\sigma^2 S_K + \frac{1}{\sqrt{K}}\right). \tag{10}$$

The proof of Theorem 3 is deferred to Appendix G.2. Theorem 3 implies that $\sigma^2 S_K$ in the right-hand side (RHS) of (9) can not be improved and therefore the lower bound of the asymptotic error. It demonstrates that the presence of asymptotic error is intrinsic and not an artifact of the convergence analysis. The nonzero asymptotic error also reveals the fundamental performance limits of using `Analog SGD` for analog training, pointing out a new venue for algorithmic development.

## 4 Eliminating Asymptotic Error of Analog Training: `Tiki-Taka`

Building upon our understanding on the modeling of gradient-based analog training in Section 2 and the asymptotic error of `Analog SGD` in Section 3, this section will be devoted to understanding means to overcome such asymptotic error in analog training.

We will focus on our study on a heuristic algorithm `Tiki-Taka` that has shown great promise in practice [22]. The key idea of `Tiki-Taka` is to maintain an auxiliary array to estimate the true gradient. To be specific, `Tiki-Taka` introduces another analog device, $P_k$, besides the main one, $W_k$. At the initialization phase, $P_k$ is initialized as 0. At iteration $k$, the stochastic gradient is computed using the main device $W_k$ and is first applied to $P_k$. Since $P_k$ is also an analog device, the change of $P_k$ still follows the dynamic (8) by replacing $W_k$ with $P_k$ and $\Delta W$ with $\nabla f(W_k) + \varepsilon_k$, that is

$$P_{k+1} = P_k + \beta(\nabla f(W_k) + \varepsilon_k) - \frac{\beta}{\tau}|\nabla f(W_k) + \varepsilon_k| \odot P_k \tag{TT-P}$$

where $\beta$ is a learning rate. After that, the value $P_{k+1}$ is read and transferred to the main array $W_k$ via

$$W_{k+1} = W_k - \alpha P_{k+1} - \frac{\alpha}{\tau}|P_{k+1}| \odot W_k. \tag{TT-W}$$

`Tiki-Taka` performs recursion (TT-P) and (TT-W) alternatively until it converges. Empirically, `Tiki-Taka` outperforms `Analog SGD` in terms of the final accuracy. However, `Tiki-Taka` is a heuristic algorithm, and there are no convergence guarantees so far. In this section, we demonstrate that the improvement of `Tiki-Taka` stems from its ability to eliminate asymptotic errors.

**Stability of `Tiki-Taka`.** As explained in Section 3, the gradient noise contributes to the asymptotic error of `Analog SGD`. To eliminate the error, the idea under `Tiki-Taka` is to reduce the noise impact. To see how `Tiki-Taka` reduces the noise, consider the case where $W_k$ is already a critical point and $P_k$ is initialized as 0, i.e. $\nabla f(W_k) = P_k = 0$. After one iteration, the weight $W_k$ drifts because of the noise. For `Analog SGD`, the expected drift is

$$\mathbb{E}[W_{k+1}] - W_k = -\alpha\mathbb{E}[\varepsilon_k] - \frac{\alpha}{\tau}\mathbb{E}[|\varepsilon_k|] \odot W_k = -\frac{\alpha}{\tau}\mathbb{E}[|\varepsilon_k|] \odot W_k \propto \alpha. \tag{11}$$

In contrast, `Tiki-Taka` updates the auxiliary array by $P_{k+1} = P_k + \beta\varepsilon_k - \frac{\beta}{\tau}|\varepsilon_k| \odot P_k = \beta\varepsilon_k$, which implies $\mathbb{E}[P_{k+1}] = 0$ and $\mathbb{E}[|P_{k+1}|] = \beta\mathbb{E}[|\varepsilon_k|]$. After the transfer, its expected drift is

$$\mathbb{E}[W_{k+1}] - W_k = -\alpha\mathbb{E}[P_{k+1}] - \frac{\alpha}{\tau}\mathbb{E}[|P_{k+1}|] \odot W_k = -\frac{\alpha\beta}{\tau}\mathbb{E}[|\varepsilon_k|] \odot W_k \propto \alpha\beta. \tag{12}$$

Comparing (12) with (11), it can be observed that `Tiki-Taka` improves the expected drift from $O(\alpha)$ to $O(\alpha\beta)$. With sufficiently small $\beta$, `Tiki-Taka` controls the drift and makes the weight stay at the critical point. For a more generic scenarios, $P_k \neq 0$. However, it is worth noting that

$$\mathbb{E}[P_{k+1}] = \mathbb{E}[P_k + \beta\varepsilon_k - \frac{\beta}{\tau}|\varepsilon_k| \odot P_k] = \left(1 - \frac{\beta}{\tau}\mathbb{E}[|\varepsilon_k|]\right) \odot P_k \tag{13}$$

which makes $P_k$ back to 0 when $\mathbb{E}[|\varepsilon_k|] \neq 0$. Therefore, by controlling the drift, the `Tiki-Taka` algorithm manages to stay near a critical point.

Note that from (13), the stability of $P_k$ relies on the presence of noise, i.e. $\mathbb{E}[|\varepsilon_k|] \neq 0$. In addition to the upper bound on the noise in Assumption 3, a lower bound for the noise is also assumed.

**Assumption 5** (Coordinate-wise i.i.d. and non-zero noise). *For any $k \geq 0$ and $i, j \in \mathcal{I}$, $[\varepsilon_k]_i$ and $[\varepsilon_k]_j$ are i.i.d. from a distribution $\mathcal{D}_c$ which ensures $\mathbb{E}_{[\varepsilon_k]_i \sim \mathcal{D}_c}[\varepsilon_k] = 0$. Furthermore, there exists $\sigma > 0$ and $c > 0$ such that $\mathbb{E}_{[\varepsilon_k]_i \sim \mathcal{D}_c}[[\varepsilon_k]_i^2] \leq \sigma^2/D$ and $\mathbb{E}_{[\varepsilon_k]_i \sim \mathcal{D}_c}[|g + [\varepsilon_k]_i|] \geq c\sigma$ for any $g \in \mathbb{R}$.*

Intuitively, Assumption 5 requires the non-zero noise, which is mild since the random sampling and the physical properties of analog devices always introduce noise. The factor $D$ in the denominator makes it consistent with Assumption 3. We discuss this assumption in more detail in Appendix D.2.

**Theorem 4** (Convergence of `Tiki-Taka`). *Suppose Assumption 1-5 hold and the learning rate is set as $\alpha = O(1/\sqrt{\sigma^2 K})$, $\beta = 8\alpha L$. It holds for Tiki-Taka that the expected infinity norm $P_k$ is upper bounded by $\mathbb{E}[\|P_{k+1}\|_\infty^2] \leq P_{\max}^2 := \frac{41L^2\tau^4 D}{c^2\sigma^2}$. Furthermore, if $\sigma^2$ and $D$ are sufficiently large so that $33P_{\max}^2/\tau^2 < 1$ it is valid that*

$$\frac{1}{K}\sum_{k=0}^{K-1}\mathbb{E}[\|\nabla f(W_k)\|^2] \leq O\left(\sqrt{\frac{(f(W_0) - f^*)\sigma^2 L}{K}}\frac{1}{1 - 33P_{\max}^2/\tau^2}\right). \tag{14}$$

The proof of Theorem 4 is deferred to Appendix H. Theorem 4 first provides the upper bound for the maximum magnitude of $P_k$ that decreases as the variance $\sigma^2$ increases or $\tau$ decreases. This observation is consistent with (13), which implies the $P_k$ tends to zero when $\frac{\beta}{\tau}\mathbb{E}[|\varepsilon_k|] \neq 0$. Ensuring stability during the training requires the noise to be sufficiently large to render the saturation degree of $P_{\max}/\tau$ sufficiently small. In addition, the condition $33P_{\max}^2/\tau^2 < 1$ requires the $D$ to be sufficiently large, which is easy to meet when training large models.

**Convergence rate.** Theorem 4 claims that `Tiki-Taka` converges at the rate $O(\sqrt{\frac{\sigma^2 L}{K}}\frac{1}{1-33P_{\max}^2/\tau^2})$. Therefore, we reach the conclusion that $\limsup_{K\to\inf}\frac{1}{K}\sum_{k=0}^{K-1}\mathbb{E}[\|\nabla f(W_k)\|^2] = 0$ and `Tiki-Taka` eliminates the asymptotic error of `Analog SGD`. Furthermore, `Tiki-Taka` improves the factor $1/(1 - W_{\max}^2/\tau^2)$ in `Analog SGD`'s convergence (c.f. (9)) to $1/(1 - 33P_{\max}^2/\tau^2)$, wherein $W_{\max}^2/\tau^2$ and $P_{\max}^2/\tau^2$ are the saturation degrees in fact. Notice that $P_k$ tends to 0 as indicated

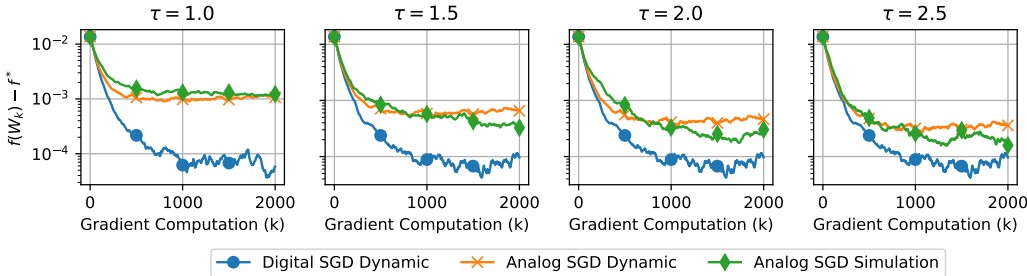

Figure 3: The convergence of digital SGD dynamic (2), analog dynamic (3) (proposed) and `Analog` SGD implemented by AIHWKIT (real behavior) under different $\tau$.

by (13) while $W_{\max}$ does not because 0 is usually not a critical point. Therefore, it usually has $P_{\max}^2/\tau^2 \ll W_{\max}^2/\tau^2$ in practice, implying faster convergence. The convergence matches the lower bound $O(\sqrt{\sigma^2 L/K})$ for general stochastic non-convex smooth optimization [24] up to a constant.

## 5  Numerical Simulations

In this section, we verify the main theoretical results by simulations on both synthetic datasets and real datasets. We use the PYTORCH to generate the curves for SGD in the simulation and use open source toolkit AIHWKIT [27] to simulate the behaviors of `Analog` SGD; see `github.com/IBM/aihwkit`. Each simulation is repeated three times, and the mean and standard deviation are reported. More details can be referred to in Appendix I. The code of our simulation implementation is available at `github.com/Zhaoxian-Wu/analog-training`.

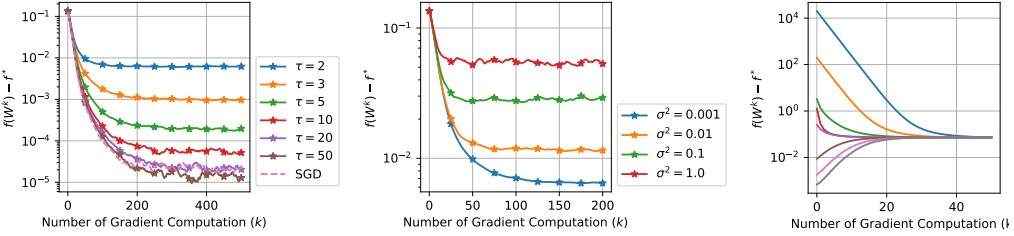

Figure 4: **(Left)** The convergence of `Analog` SGD under different $\tau$. Reducing $\tau$ leads to a decrease in asymptotic error. When $\tau$ is sufficiently large, `Analog` SGD tends to have a similar performance to digital SGD. **(Middle)** The convergence of `Analog` SGD on noise devices under different $\sigma^2$. **(Right)** `Analog` SGDs that are initialized to different places converge to the same error.

### 5.1  Verification of the analog training dynamic

To verify that the proposed dynamic (3) characterizes analog training better than SGD dynamic (2), we conduct a numerical simulation on a least-squares task, and compare `Analog` SGD implemented by AIHWKIT, the digital and analog dynamics given by (2) and (3), respectively; see Figure 3. The results show that the proposed dynamic provides an accurate approximation of `Analog` SGD.

### 5.2  Ablation study on the asymptotic training error

We verify some critical claims about the asymptotic error of `Analog` SGD on a least-squares task.

**Impact of** $\tau$. To verify Theorem 2 that the error diminishes with a larger $\tau$, we assign a range of value $\tau$ and plot the convergence of `Analog` SGD. The result is reported on the Left of Figure 4. When $\tau$ is small, the asymmetric bias introduces a notable gap between Analog and Digital SGD. As $\tau$ increases, the gap diminishes. The result demonstrates the asymptotic error decreases as $\tau$ increases.

**Impact of** $\sigma^2$. To verify the asymptotic error is proportional to $\sigma^2$, we inject noise with different variances. The result is reported in the middle of Figure 4. The result illustrates that the asymptotic error increases as the noise increases.

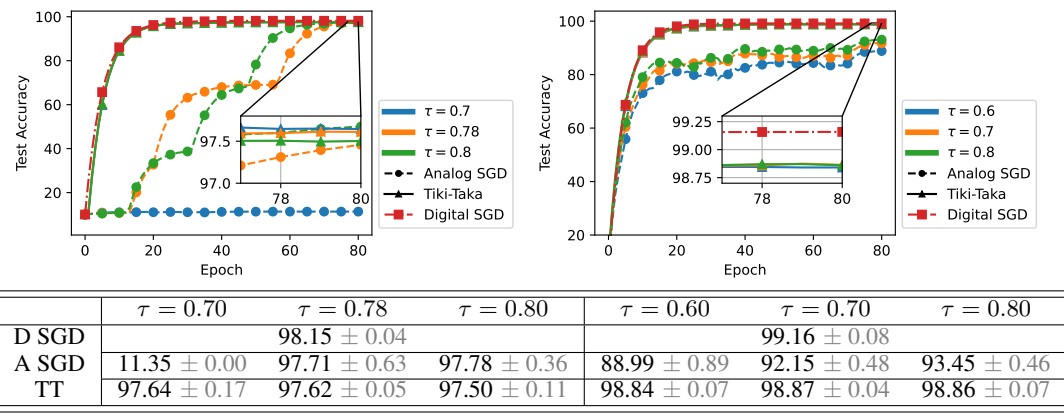

| | $\tau = 0.70$ | $\tau = 0.78$ | $\tau = 0.80$ | $\tau = 0.60$ | $\tau = 0.70$ | $\tau = 0.80$ |
|---|---|---|---|---|---|---|
| D SGD | | $98.15 \pm 0.04$ | | | $99.16 \pm 0.08$ | |
| A SGD | $11.35 \pm 0.00$ | $97.71 \pm 0.63$ | $97.78 \pm 0.36$ | $88.99 \pm 0.89$ | $92.15 \pm 0.48$ | $93.45 \pm 0.46$ |
| TT | $97.64 \pm 0.17$ | $97.62 \pm 0.05$ | $97.50 \pm 0.11$ | $98.84 \pm 0.07$ | $98.87 \pm 0.04$ | $98.86 \pm 0.07$ |

Figure 5: The test accuracy curves and tables for the model training. "D SGD", "A SGD", and "TT" represent Digital SGD, `Analog SGD` and `Tiki-Taka`, respectively; **(Left)** FCN. **(Right)** CNN.

**Impact of the initialization**. To demonstrate the asymptotic error is not artificial, we perform `Analog SGD` from different initializations. The result is reported on the right of Figure 4. The result illustrates that `Analog SGD` converges to a similar location regardless of the initialization. The smooth convergence curve ensures the error comes from bias instead of limited machine precision. Therefore, the asymptotic error is intrinsic and independent of the initialization.

### 5.3 Analog training performance on real dataset

We also train vision models to perform image classification tasks on real datasets.

**MNIST FCN/CNN.** We train Fully-connected network (FCN) and convolution neural network (CNN) models on MNIST dataset and see the performance of `Analog SGD` and `Tiki-Taka` under various $\tau$. The results are reported in Figure 5. By reducing the variance, `Tiki-Taka` outperforms `Analog SGD` and reaches comparable accuracy with digital SGD. On both

| | Resnet18 | Resnet34 | Resnet50 |
|---|---|---|---|
| Digital SGD | 93.03 | 93.44 | 95.92 |
| Analog SGD | 93.58 | 93.58 | 95.51 |
| Tiki-Taka | 93.74 | 95.15 | 95.54 |

Table 2: The test accuracy of Resnet training on CIFAR10 dataset after 100 epochs.

of the architectures, the accuracy of `Tiki-Taka` drops by $< 1\%$. In the FCN training, `Analog SGD` achieves acceptable accuracy on $\tau = 0.78$ and $\tau = 0.80$ but converges much more slowly. In the CNN training, the accuracy of `Analog SGD` always drops by $> 6\%$.

**CIFAR10 Resnet.** We also train three Resnet models with different sizes on CIFAR10 dataset. The last layer is replaced by a fully-connected layer mapped onto an analog device with parameter $\tau = 0.8$. The results are shown in Table 2. In this task, `Analog SGD` does not suffer from a significant accuracy drop but is still worth that `Tiki-Taka`, A surprising observation for analog training is that both `Analog SGD` and `Tiki-Taka` outperform Digital SGD. We conjecture it happens because the noise introduced by analog devices makes the network more robust to outlier data.

## 6 Conclusions and Limitations

This paper points out that `Analog SGD` does not follow the dynamic of digital SGD and hence, we propose a better dynamic to formulate the analog training. Based on this dynamic, we studies the convergence of two gradient-based analog training algorithms, `Analog SGD` and `Tiki-Taka`. The theoretical results demonstrate that `Analog SGD` suffers from asymptotic error, which comes from the noise and asymmetric update. To overcome this issue, we show that `Tiki-Taka` is able to stay in the critical point without suffering from an asymptotic error. Numerical simulations demonstrate the existence of `Analog SGD`'s asymptotic error and the efficacy of `Tiki-Taka`. One limitation of this work is that the current analysis is device-specific that applies to asymmetric linear device. While it is an interesting and popular analog device, it is also important to extend our convergence analysis to more general analog devices and develop other device-specific analog algorithms in future work.

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

# Supplementary Material for "Towards Exact Gradient-based Training on Analog In-memory Computing"

## Table of Contents

## A  Literature Review

This section briefly reviews literature that is related to this paper, as complementary to Section 1.

**Inference and other applications of analog devices.** Before designing hardware for training, a series of AIMC prototypes focus on accelerating the inference phase [29, 30, 31, 32, 33, 12, 34]. The state-of-the-art result shows that the analog inference is capable of reaching comparable accuracy with digital inference [28]. Besides analog model training and inference, researchers also managed to exploit the advantages of analog devices to facilitate the machine learning task. For example, energy-based learning algorithms have been studied in [35], and the neural architecture search on analog devices has been studied in [36].

**Relation with physical neural network.** Our work is also in spirit related to the recent works on physical neural networks (PNNs) [37, 38]. But we are particularly interested in AIMC hardware with resistive crossbar arrays, while PNN is a generic concept of implementing neural networks via a

physical system. Various hardware is capable of supporting PNN, such as holographic grating [39], wave-based systems [40], and photonic networks [41], to name a few.

**Convergence analysis of gradient-based algorithms.** In the digital domain, a series of literature has discussed the convergence of gradient-based algorithms. As the basis of neural model training, much work focuses on the convergence of SGD [42, 43, 44, 23] and its variety stochastic gradient descent with momentum (SGDM) [45, 46, 47, 48]. Notice that the difference between digital SGD (2) and `Analog SGD` (3) lies on the asymmetric bias term, which implies `Analog SGD` can be regarded as a digital SGD with bias. From this perspective, the convergence of `Analog SGD` (c.f. Theorem 2) is similar to that of the digital bias-SGD counterparts. However, the results in this paper are still challenging and non-trivial, especially in the convergence of `Tiki-Taka`, because the updates of both $W_k$ and $P_k$ involve asymmetric bias.

Nowadays, gradient-based algorithms demonstrate their powerful capabilities in model training. In fact, both SGD and SGDM have reached the theoretical lower bound of the convergence rate, $\sqrt{\sigma^2/K}$ [24]. This lower bound is a generic result for any stochastic gradient-based algorithms, which means that the convergence of gradient-based analog algorithms, including `Analog SGD` and `Tiki-Taka`, is also subject to it.

# B   Implementation of Asymmetric Updated by Pulse Update

The proposed dynamic (3) of `Analog SGD` in Section 2 is inspired by the empirical studies in [22] and [19]. However, the proposed dynamic (3) is only an approximation because we can not directly apply an arbitrary increment $\Delta w$ on the weights. Instead, we need to send a series of *pulses* to analog devices, which leads to the change on the weight. In this section, we introduce the implementation of analog update $U(\cdot, \cdot)$ in (6) by pulse update and analyze the error of pulse update implementation.

**Implementation.** Similar to $U(\cdot, \cdot)$, we introduce the update function $U_p(\cdot, \cdot) : \mathbb{R} \times \{+, -\} \to \mathbb{R}$ for the pulsed update, defined by

$$U_p(w, s) := w + \Delta w_{\min} \cdot q_s(w) = \begin{cases} w + \Delta w_{\min} \cdot q_+(w), & s = +, \\ w - \Delta w_{\min} \cdot q_-(w), & s = -, \end{cases} \tag{15}$$

where $\Delta w_{\min} > 0$ is the *response step size* determined by devices. Since $\Delta w_{\min}$ is the minimum change of weight, it is sometimes called the *resolution* or *granularity* of the devices. Given the initial weight $w$, the updated weight after receiving one pulse is $U_p(W_k, s)$ where the update sign $s = +$ if $\Delta w \geq 0$ and $s = -$ otherwise.

The response step size $\Delta w_{\min}$ is usually known by physical measurement. In order to approximate $U(w, \Delta w)$, analog device first computes the *pulse series length* (bit length) by

$$\text{BL} := \left\lceil \frac{|\Delta w|}{\Delta w_{\min}} \right\rceil, \tag{16}$$

which ensures

$$|\,\text{BL}\,\Delta w_{\min} - |\Delta w|\,| \leq \Delta w_{\min} \quad \text{or} \quad |s\,\text{BL}\,\Delta w_{\min} - \Delta w| \leq \Delta w_{\min}. \tag{17}$$

After that, a total of BL pulses are sent to the analog device, forcing the weight to become

$$U(w, \Delta w) \approx \underbrace{U_p \circ U_p \circ \cdots \circ U_p}_{\times \,\text{BL}}(w, s) = U_p^{\text{BL}}(w, s). \tag{18}$$

which implement the analog update $U(w, \Delta w)$.

**Error analysis.** Directly expending the (18) yields

$$U_p^{\text{BL}}(w, s) = w + s\Delta w_{\min} \sum_{t=0}^{\text{BL}-1} q_s(w + t\Delta w_{\min} \cdot \text{sign}(\Delta w)) \tag{19}$$

$$= w + s\Delta w_{\min} \sum_{t=0}^{\text{BL}-1} \Big( q_s(w) + t\Delta w_{\min} \cdot q_s'(w) \cdot \text{sign}(\Delta w) \Big) + o(\Delta w_{\min})$$

$$= w + s\Delta w_{\min} \operatorname{BL} q_s(w) + (\Delta w_{\min})^2 \frac{\operatorname{BL}(\operatorname{BL}-1)}{2} q_s'(w) \cdot \operatorname{sign}(\Delta w) + o(\Delta w_{\min})$$

$$= w + \Delta w \cdot q_s(w) + O(\Delta w_{\min}) + O((\Delta w)^2),$$

which indicates the difference between $U(w, \Delta w)$ and $U_p^{\mathrm{BL}}(w, s)$ are higher order infinitesimal. As a concrete example, the $\Delta w_{\min}$ is set as 0.002 [49] or 0.0949 [20] for ReRAM devices.

For asymmetric linear device (ALD, defined in Section 2), it holds that $|q_+'(w)| = |q_-'(w)| = \frac{1}{\tau}$ and $q_+''(w) = q_-''(w) = 0$. Consequently, the difference between $U(w, \Delta w)$ and $U_p^{\mathrm{BL}}(w, s)$ is

$$|U_p^{\mathrm{BL}}(w, s) - U(w, \Delta w)| \leq \frac{\Delta w_{\min}}{\tau} + (\Delta w_{\min})^2 \frac{\operatorname{BL}(\operatorname{BL}-1)}{2\tau} \leq \frac{\Delta w_{\min}}{\tau} + \frac{(\Delta w)^2}{2\tau}. \qquad (20)$$

Since $\Delta w_{\min}$ and $\Delta w = O(\alpha)$ are usually small, the error can usually be ignored.

**Significance.** The significance of the analysis in this section is that it explains how to estimate the response factors $q_+(\cdot)$ and $q_-(\cdot)$ in (3). In (15), it shows that they are, in fact, the response factors of the given devices, which can be measured by physic measures [49]. In Section C, we showcase various capable base materials for analog devices and their response factors.

## C   Examples of Analog Devices

In AIMC accelerators, weights of models can be represented by the conductance of based materials, such as PCM [50, 51], ReRAM [52, 53], CBRAM [54, 55], or ECRAM [56]. In this section, we showcase a spectrum of alternative devices for analog training.

### C.1   Example 1: Linear step device (Soft bounds devices)

A large range of devices, including ReRAM [49, 20], Capacitor [57], EcRAM [57, 56], EcRamMO [58], have linear response factor

$$q_+(w) = 1 - \frac{w - w^{\diamond}}{\tau_{\max}}, \quad q_-(w) = 1 - \frac{w - w^{\diamond}}{\tau_{\min}}, \qquad (21)$$

where $\tau_{\min} < 0 < \tau_{\max}$ and $w^{\diamond}$ are constants, whose meaning will be explained later. With these response factors, the function $F(\cdot)$ and $G(\cdot)$ can be written as

$$F(w) = 1 - \frac{1}{2}(\frac{1}{\tau_{\max}} + \frac{1}{\tau_{\min}})(w - w^{\diamond}), \qquad (22)$$

$$G(w) = -\frac{1}{2}(\frac{1}{\tau_{\max}} - \frac{1}{\tau_{\min}})(w - w^{\diamond}). \qquad (23)$$

ALD used in the paper is a special linear step device with $\tau_{\max} = \tau$ and $\tau_{\min} = -\tau$.

### C.2   Example 2: Power step device

For power step device, $q_+(w)$ and $q_-(w)$ are power function with respect to $W$, which models implements synapse model [59, 60].

$$q_+(w) = \left(\frac{\tau_{\max} - w}{\tau_{\max} - \tau_{\min}}\right)^{\gamma^+}, \quad q_-(w) = \left(\frac{w - \tau_{\min}}{\tau_{\max} - \tau_{\min}}\right)^{\gamma^-} \qquad (24)$$

where $\gamma^+$ and $\gamma^-$ are parameters determined by materials.

### C.3   Example 3: Exponential step device

In some ReRAM and CMOS-like devices [61, 21], the response factors are captured by an exponential function, i.e. Exponential update step or CMOS-like update behavior.

$$q_+(w) = 1 - \exp\left(-\gamma^+ \cdot \frac{\tau_{\max} - w}{\tau_{\max} - \tau_{\min}}\right), \qquad (25)$$

$$q_-(w) = 1 - \exp\left(-\gamma^- \cdot \frac{w - \tau_{\min}}{\tau_{\max} - \tau_{\min}}\right). \qquad (26)$$

In Figure 6, the curve of response factors and $F(\cdot)$ and $G(\cdot)$ are illustrated.

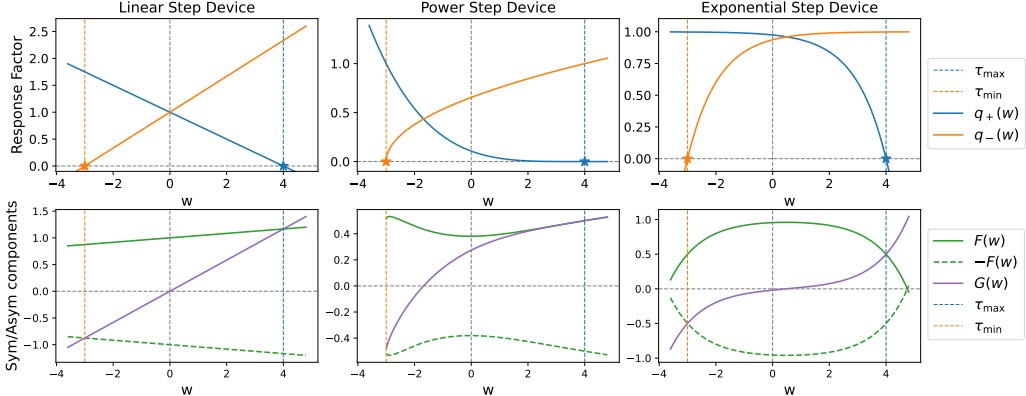

Figure 6: Response factors for different devices (without zero-shifting).

# D  Verification of Assumptions

## D.1  Verification of Assumption 4 under Strongly Convex Objective

This section proves the weight $W_k$ is strongly bounded when the loss function is strongly convex. We prove the result for both Analog GD ($\varepsilon_k = 0$) and `Analog SGD` with $\varepsilon_k$ possessing some special structure. Before proving this, we introduce strongly convex assumption.

**Assumption 6** ($\mu$-strongly convex). *The objective $f(W)$ is $\mu$-strongly convex, i.e.*

$$\|\nabla f(W) - \nabla f(W')\| \geq \mu\|W - W'\|, \quad \forall W, W' \in \mathbb{R}^D. \tag{27}$$

Assumption 6 ensures the critical point $W^*$ (and hence optimal point) is unique.

### D.1.1  Bounded Saturation of Analog GD

The following theorem provides a sufficient condition under which the Assumption 4 holds.

**Theorem 5** (Bounded saturation of Analog GD). *Suppose Assumptions 1 and 6 hold and define*

$$W_{\max} := \frac{L}{\mu}\|W_0 - W^*\| + \|W^*\|_\infty. \tag{28}$$

*The sequence $\{W_k\}$ generated by Analog GD satisfies $\|W_k\|_\infty \leq W_{\max}$.*

Theorem 5 claims that if both $\frac{L}{\mu}\|W_0 - W^*\|$ and $\|W^*\|_\infty$ are sufficiently small, which means the optimal point $W^*$ is located inside the active region and $W_0$ is properly initialized, the saturation degree $\|W_k\|_\infty/\tau$ can be sufficiently small as well, even though no projection operation is applied.

*Proof of Theorem 5.* Given $f(\cdot)$ is $L$-smooth (Assumption 1), we know from (57) that

$$f(W_{k+1}) \leq f(W_k) - \frac{\alpha}{2}\left(1 - \frac{\|W_k\|_\infty^2}{\tau^2}\right)\|\nabla f(W_k)\|^2 \leq f(W_k) \tag{29}$$

where the second inequality comes from the bounded weight (Theorem 1). Consequently, we reach the conclusion that $f(W_k) \leq f(W_0), \forall k \in \mathbb{N}$. The strongly convex assumption (Assumption 6) claims that

$$\|W_k - W^*\| \leq \frac{1}{\mu}(f(W_k) - f^*) \leq \frac{1}{\mu}(f(W_0) - f^*) \leq \frac{L}{\mu}\|W_0 - W^*\|. \tag{30}$$

Therefore, triangle inequality ensures that

$$\|W_k\|_\infty \leq \|W_k - W^*\|_\infty + \|W^*\|_\infty \tag{31}$$
$$\leq \|W_k - W^*\| + \|W^*\|_\infty$$
$$\leq \frac{L}{\mu}\|W_0 - W^*\| + \|W^*\|_\infty \tag{32}$$

which completes the proof. $\qquad\square$

**Remark 2** (Bounded weight of digital GD). *Theorem 5 also holds for digital GD, which can be achieved by forcing $\tau$ in inequality (29) to infinite.*

In high-dimensional cases ($D > 1$), Theorem 5 does not exclude the situation $\|W_k\|_\infty > \|W^*\|_\infty$, which is unfavorable in the example of `Analog SGD`'s lower bound (c.f. Section G.2). To facilitate the proof of the lower bound, we consider the scalar case ($D = 1$). To distinguish between vectors and scalars, we use the notation $w_k$ (and $w^*$) instead of $W_k$ (and $W^*$) in the next theorem.

**Theorem 6** (Bounded saturation of Analog GD with scaler weight). *Suppose Assumptions 1 and 6 hold and $w_k \in \mathbb{R}$ is a scalar. If the learning rate $\alpha \leq \frac{1}{2L}$, the following statements hold true*

*(Case 1) If $w_0 \geq w^*$, $w_0 \geq w_k \geq (1 - \alpha\mu(1 - \frac{\|w_k\|_\infty}{\tau}))(w_k - w^*) + w^* \geq w_{k+1} \geq w^*$;*

*(Case 2) If $w_0 < w^*$, $w_0 \leq w_k \leq (1 - \alpha\mu(1 - \frac{\|w_k\|_\infty}{\tau}))(w_k - w^*) + w^* \leq w_{k+1} \leq w^*$.*

*Consequently, the sequence $\{w_k\}$ generated by Analog GD satisfies the following relation $|w_k| \leq w_{\max} := \min\{|w_0|, |w^*|\}$.*

Theorem 6 asserts that $\{w_k\}$ converges monotonically from $w_0$ to $w^*$. If $|w_0| \leq |w^*|$, we have $w_{\max} = |w^*|$, which improves the bound in Theorem 5.

*Proof of Theorem 6.* Expanding the objective $f(\cdot)$ in a Taylor series around $w$ yields that[2]

$$f(w^*) = f(w) + \langle \nabla f(w), w - w^* \rangle + \tilde{\mu}(w)\|w - w^*\|^2, \tag{33}$$

where $\tilde{\mu}(w)$ depends on $w$ and satisfies $\mu \leq \tilde{\mu}(w) \leq L$ because of the smoothness (Assumption 1) and strong convexity (Assumption 6) assumptions. Consequently, the gradient on $w$ can be written as

$$f(w) = \tilde{\mu}(w)(w - w^*) \tag{34}$$

with which the gradient descent has the following property

$$w_k - \alpha\nabla f(w_k) - w^* = (1 - \alpha\tilde{\mu}(w_k))(w_k - w^*). \tag{35}$$

Using this inequality and manipulating $\|w_{k+1} - w^*\|^2$ as

$$\|w_{k+1} - w^*\|^2 \tag{36}$$

$$= \|w_k - \alpha\nabla f(w_k) - \frac{\alpha}{\tau}|\nabla f(w_k)| \odot w_k - w^*\|^2$$

$$= \|w_k - \alpha\nabla f(w_k) - w^*\|^2 + \frac{2\alpha}{\tau}\langle w_k - \alpha\nabla f(w_k) - w^*, |\nabla f(w_k)| \odot w_k\|\rangle$$

$$+ \frac{\alpha^2}{\tau^2}\||\nabla f(w_k)| \odot w_k\|^2$$

$$\overset{(a)}{\leq} \left(1 - 2\alpha\tilde{\mu}(w_k) + \alpha^2\tilde{\mu}(w_k)^2 + 2\alpha\tilde{\mu}(w_k)(1 - \alpha\tilde{\mu}(w_k))\frac{\|w_k\|_\infty}{\tau} + \frac{\alpha^2\tilde{\mu}(w_k)^2\|w_k\|_\infty^2}{\tau^2}\right)\|w_k - w^*\|^2$$

$$= \left(1 - 2\alpha\tilde{\mu}(w_k)(1 - \frac{\|w_k\|_\infty}{\tau}) + \alpha^2\tilde{\mu}(w_k)^2(1 - 2\frac{\|w_k\|_\infty}{\tau} + \frac{\|w_k\|_\infty^2}{\tau^2})\right)\|w_k - w^*\|^2$$

$$= \left(1 - \alpha\tilde{\mu}(w_k)(1 - \frac{\|w_k\|_\infty}{\tau})\right)^2\|w_k - w^*\|^2$$

where the $(a)$ uses Cauchy's inequality. Noticing Theorem 1 claims $\frac{\|w_k\|_\infty}{\tau} \leq 1$ and the learning rate is chosen as $\alpha\tilde{\mu}(w_k) \leq \alpha L \leq 1$, we have $\|w_{k+1} - w^*\|^2 \leq \|w_k - w^*\|^2$.

In addition, Lemma 1 claims that

$$\|w_{k+1} - w_k\| \leq \|1 + w_k/\tau\|\|\alpha\nabla f(w_k)\| \leq 2\alpha L\|w_k - w^*\| \leq \|w_k - w^*\| \tag{37}$$

where the second inequality holds because Theorem 1 and smoothness assumption (Assumption 1 guarantee $\|1 + w_k/\tau\| \leq 2$ and $\|\nabla f(w_k)\| \leq \|w_k - w^*\|$, respectively. This fact implies that

---

[2]Notice that the $\ell_2$-norm $\|\cdot\|$ and $\ell_\infty$-norm $\|\cdot\|_\infty$ reduce to absolute value, when $w$ is a scalar. Similarly, inner product $\langle\cdot,\cdot\rangle$ reduces to multiplication. We adopt the same notations for both vector and scalar weights to make the notations consistent and easy to read.

$w_{k+1} - w^*$ has the same sign with $w_k - w^*$. Consequently, inequality (36) and (37) show that in case 1, if $w_k \geq w^*$, we have

$$w_k \geq (1 - \alpha\mu(1 - \frac{\|w_k\|_\infty}{\tau}))(w_k - w^*) + w^* \tag{38}$$

$$\geq w_{k+1} \geq w^*.$$

Keeping using this inequality reaches the result. Case 2 can be proved by the similar method. □

### D.1.2 Bounded Saturation of `Analog SGD`

The following theorem provides a sufficient condition under which the Assumption 4 holds. In the section, we show that if the noise has special structures, the saturation degree $\|W_k\|_\infty / \tau$ is bounded.

**Assumption 7** (Random sample noise). *The noise can be written as $\varepsilon = \nabla f(W; \xi) - \nabla f(W)$, where $\xi$ is a random variable sampled from an underline data distribution $\mathcal{D}_f$, $f(W; \xi)$ is a function of $W$ and $\xi$, and the gradient $\nabla f(W; \xi)$ is taken over $W$.*

Assumption 7 holds if the noise comes from the random sampling from a series of functions.

**Theorem 7** (Bounded saturation of `Analog SGD`). *Suppose Assumption 7 holds and each $f(W; \xi)$ is L-smooth (Assumptions 1) and $\mu$-strongly convex (Assumptions 6) with respect to $W$. Denote the minimum of $f(W; \xi)$ as $W_\xi^*$. If the supremum $\sup_\xi\{\|W_\xi^*\|_\infty\}$ exists, define*

$$W_{\max} := \frac{L}{\mu} \sup_\xi\{\|W_0 - W_\xi^*\|\} + \sup_\xi\{\|W_\xi^*\|_\infty\}. \tag{39}$$

*The sequence $\{W_k\}$ generated by `Analog SGD` satisfies $\|W_k\|_\infty \leq W_{\max}$ uniformly.*

Given the strong convexity with respect to $W$, $f(W; \xi)$ has unique minimum, and hence $W_\xi^*$ is well-defined. Similarly, the supremum $\sup_\xi\{\|W_0 - W_\xi^*\|\}$ exists since $\sup_\xi\{\|W_\xi^*\|_\infty\}$ exists. Theorem 7 claims that the saturation degree is bounded given the noise comes from random sampling over a series of smooth and strongly convex functions. Similar to Theorem 5, Theorem 7 also holds for digital GD by forcing $\tau$ in inequality (29) to infinite. The proof of Theorem 7 is inspired by that of [62, Lemma 1].

*Proof.* According to Theorem 5, it holds that

$$\|W_k - W_\xi^*\| \leq \frac{L}{\mu}\|W_0 - W_\xi^*\|. \tag{40}$$

Therefore, triangle inequality ensures that

$$\|W_k\|_\infty \leq \sup_\xi\{\|W_k - W_\xi^*\|_\infty\} + \sup_\xi\{\|W_\xi^*\|_\infty\} \tag{41}$$

$$\leq \sup_\xi\{\|W_k - W_\xi^*\|\} + \sup_\xi\{\|W_\xi^*\|_\infty\}$$

$$\leq \frac{L}{\mu} \sup_\xi\{\|W_0 - W_\xi^*\|\} + \sup_\xi\{\|W_\xi^*\|_\infty\}$$

which completes the proof. □

Similar to Theorem 6, we can improve the bound when the weight is scalar ($D = 1$). The notation $w_k$ (and $w^*$) instead of $W_k$ (and $W^*$) are adopted here to indicate the scalar situation.

**Theorem 8** (Bounded saturation of Analog GD). *Suppose Assumption 7 holds and for any $\xi$, $f(w; \xi)$ is L-smooth (Assumptions 1) and $\mu$-strongly convex (Assumptions 6) with respect to $w$. Denote the minimum of $f(w; \xi)$ as $w_\xi^*$. If the supremum $\sup_\xi\{\|w_\xi^*\|\}$ exists, it holds for any $k \in \mathbb{N}$ that*

$$\min\left\{w_0, \inf_\xi\{w_\xi^*\}\right\} \leq w_k \leq \max\left\{w_0, \sup_\xi\{w_\xi^*\}\right\}. \tag{42}$$

*Proof of Theorem 8.* The theorem is proved by induction. The statement holds trivially at $k = 0$. Suppose (42) holds for $k$. If $W_k \geq w^*$, Theorem 6 guarantees that $w_k \geq w_{k+1} \geq w_\xi^*$ and hence

$$\min\left\{w_0, \inf_\xi\{w_\xi^*\}\right\} \leq w_\xi^* \leq w_{k+1} \leq w_k \leq \max\left\{w_0, \sup_\xi\{w_\xi^*\}\right\}. \tag{43}$$

On the contrary, if $w_k \leq w^*$, Theorem 6 guarantees that $w_k \leq w_{k+1} \leq w_\xi^*$ and hence

$$\min\left\{w_0, \inf_\xi\{w_\xi^*\}\right\} \leq w_k \leq w_{k+1} \leq w_\xi^* \leq \max\left\{w_0, \sup_\xi\{w_\xi^*\}\right\} \tag{44}$$

which implies $w_{k+1}$ still satisfies (42). Now the conclusion is reached. $\qquad\square$

## D.2 Verification of Assumption 5: Non-zero Property of Gaussian Noise

This section verifies Assumption 5. In principle, Assumption 5 requires the expectation that noise is non-zero. To see that, note that when the probability density function of $\mathcal{D}$ is non-zero at only one point, $c$ equals zero. On the contrary, large $c$ is at the other side of the spectrum, which appears when the probability density function is relatively "uniformly distributed" throughout the probability space. In analog acceleration devices, it is mild because noise is introduced during computation.

Generally, the estimation of $c$ can be challenging because it involves the computation of the raw momentum $\mathbb{E}_\varepsilon[|g + \varepsilon|]$ of the noise. Scarifying the rigorousness a bit but providing an intuitive result, we get the parameter $c$ by considering an approximation of this raw momentum and deducing a bound for this approximation.

In this section, we demonstrate that for Gaussian noise $\varepsilon \sim \mathcal{N}(0, \sigma^2)$, $c$ can be selected as $\sqrt{\frac{2}{\pi}}$. For any $g$, it holds that

$$\mathbb{E}_\varepsilon[\|\mathbf{1} - \frac{\beta}{\tau}|g + \varepsilon|\|_\infty^2] = \mathbb{E}_\varepsilon[(1 - \frac{\beta}{\tau}|g + \varepsilon|)^2] \tag{45}$$

$$= 1 - \frac{2\beta}{\tau}\mathbb{E}_\varepsilon[|g + \varepsilon|] + \frac{\beta^2}{\tau^2}\mathbb{E}_\varepsilon[(g + \varepsilon)^2]$$

$$= 1 - \frac{2\beta}{\tau}\mathbb{E}_\varepsilon[|g + \varepsilon|] + \frac{\beta^2}{\tau^2}(g^2 + \sigma^2).$$

Because $g + \varepsilon$ is Gaussian random variable with mean $g$ and variance $\sigma^2$, the second term in the RHS has closed form [63] given by

$$\mathbb{E}_\varepsilon[|g + \varepsilon|] = \sigma\sqrt{\frac{2}{\pi}} \, {}_1F_1\left(-\frac{1}{2}; \frac{1}{2}; -\frac{g^2}{2\sigma^2}\right). \tag{46}$$

In the equation above, ${}_1F_1(\cdot; \cdot; \cdot)$ is used to denote Kummer's confluent hypergeometric functions. It has been shown [64, 13.1.5] that when $z \leq 0$, ${}_1F_1(\cdot; \cdot; \cdot)$ has the asymptotic property

$${}_1F_1(a; b; z) = \frac{\Gamma(b)}{\Gamma(b - a)}(-z)^{-a}(1 + O(|z|^{-1})), \tag{47}$$

where $\Gamma(\cdot)$ is Gamma function. When $z$ is a small, ${}_1F_1(a; b; z)$ has the following approximation for any $a, b \in \mathbb{R}$ [65, Sec 11.2]

$${}_1F_1(a; b; z) \sim 1 + z. \tag{48}$$

Let $z = -\frac{g^2}{2\sigma^2}$. On the one hand when $\sigma \sim 0$, $z$ is large and (47) with $a = -\frac{1}{2}$ and $b = \frac{1}{2}$ asserts that

$$\mathbb{E}_\varepsilon[|g + \varepsilon|] \sim \sigma\sqrt{\frac{2}{\pi}} \cdot \frac{\Gamma(1/2)}{\Gamma(1)}\sqrt{\frac{g^2}{2\sigma^2}}(1 + O(\frac{2\sigma^2}{g^2})) = \sqrt{\frac{2}{\pi}}\sigma \cdot \sqrt{\pi}(1 + \sqrt{\frac{g^2}{2\sigma^2}}). \tag{49}$$

On the other hand, when $\sigma \to \infty$, $z$ is small and (48) provides that

$$\mathbb{E}_\varepsilon[|g + \varepsilon|] \sim \sigma\sqrt{\frac{2}{\pi}} \cdot (1 + \frac{g^2}{2\sigma^2}) = \sqrt{\frac{2}{\pi}}\sigma \cdot (1 + \frac{g^2}{2\sigma^2}). \tag{50}$$

In conclusion, choosing $c = \sqrt{\frac{2}{\pi}}$ yields $\mathbb{E}_\varepsilon[|g + \varepsilon|] \gtrsim \sqrt{\frac{2}{\pi}}\sigma$. Here $\gtrsim$ is used to indicate the lower bound holds for an approximation of the left-hand side (LHS).

# E  Proof of Analog Training Properties

## E.1  Proof of Lemma 1: Saturation and Fast Reset

**Lemma 1** (Saturation and fast reset). *For ALD with a general $w^\diamond$, the following statements are valid*

*(Saturation)*   *If* $\mathrm{sign}(\Delta w) = \mathrm{sign}(w_k)$, *it holds that* $|w_{k+1} - w_k| = |1 - |w_k - w^\diamond|/\tau| \cdot |\Delta w|$.

*(Fast Reset)*   *If* $\mathrm{sign}(\Delta w) = -\mathrm{sign}(w_k)$, *it holds that* $|w_{k+1} - w_k| = |1 + |w_k - w^\diamond|/\tau| \cdot |\Delta w|$.

*Proof of Lemma 1.*  The proof can be completed by simply extending the LHS of

$$|w_{k+1} - w_k| = \left| \Delta w - \frac{1}{\tau}|\Delta w|(w_k - w^\diamond) \right| \tag{51}$$

$$= \left| |\Delta w|(\mathrm{sign}(\Delta w) - \frac{1}{\tau}(w_k - w^\diamond)) \right|$$

$$= |\mathrm{sign}(\Delta w) - \frac{1}{\tau}(w_k - w^\diamond)| \cdot |\Delta w|$$

$$= \left| 1 - \frac{\mathrm{sign}(w_k - w^\diamond)}{\mathrm{sign}(\Delta w)} \frac{|w_k - w^\diamond|}{\tau} \right| \cdot |\Delta w|.$$

Using the fact that $w^\diamond = 0$ completes the proof.  □

## E.2  Proof of Theorem 1: Bounded Weight

This section proves that the weight $W_k$ is bounded.

**Theorem 1** (Bounded weight). *Denote $\|W\|_\infty$ as the $\ell_\infty$ norm of $W$. Given $\|W_0\|_\infty \leq \tau$ and for any sequence $\{\Delta W_k : k \in \mathbb{N}\}$, which satisfies $\|\Delta W_k\|_\infty \leq \tau$, it holds that $\|W_k\|_\infty \leq \tau$, $\forall k \in \mathbb{N}$.*

*Proof of Theorem 1.*  Without causing confusion, omit the superscript $k$ and denote the $i$-th coordinate of $W_k$ and $\Delta W_k$ as $w_i$ and $g_i$, respectively. It holds that $\|W_k\|_\infty^2 < \tau$ and $|g_{ij}| \leq \tau$

$$\|W_{k+1} - W^\diamond\|_\infty^2 = \|W_k - W^\diamond + \Delta W_k - \frac{1}{\tau}|\Delta W_k| \odot (W_k - W^\diamond)\|_\infty^2 \tag{52}$$

$$= \max_{i \in \mathcal{I}}(w_i - w^\diamond - g_i - \frac{1}{\tau}|g_i|(w_i - w^\diamond))^2$$

$$= \max_{i \in \mathcal{I}} \left( (1 - \frac{1}{\tau}|g_i|)^2(w_i - w^\diamond)^2 - 2g_i(1 - \frac{1}{\tau}|g_i|)(w_i - w^\diamond) + g_i^2 \right)$$

$$\leq \max_{i \in \mathcal{I}} \left( (1 - \frac{1}{\tau}|g_i|)^2\tau^2 + 2|g_i|\,(1 - \frac{1}{\tau}|g_i|)\tau + g_i^2 \right)$$

$$= \max_{i \in \mathcal{I}} \left( (1 - \frac{1}{\tau}|g_i|)\,\tau + |g_i| \right)^2$$

$$= \tau^2.$$

Taking square root on both sides completes the proof.  □

# F  Proof of Analog Gradient Descent Convergence

In Section 3, we show that the asymptotic error is proportional to the noise variance $\sigma^2$. The following theorem prove that the Analog GD converges to a critical point.

**Theorem 9** (Convergence of Analog GD). *Under Assumption 1, 2 and 4, if the learning rate is set as $\alpha \leq \frac{1}{L}$, it holds that*

$$\frac{1}{K} \sum_{k=0}^{K-1} \|\nabla f(W_k)\|^2 \leq \frac{2(f(W_0) - f^*)L}{K} \frac{1}{1 - W_{\max}^2/\tau^2}. \tag{53}$$

A surprising conclusion of Theorem 9 is that the asymmetric update actually will not introduce any asymptotic error in training. It also indicates that the asymptotic error comes from the interaction between asymmetric update and noise from another perspective.

*Proof of Theorem 9.* The $L$-smooth assumption (Assumption 1) implies that

$$f(W_{k+1}) \leq f(W_k) + \langle \nabla f(W_k), W_{k+1} - W_k \rangle + \frac{L}{2} \|W_{k+1} - W_k\|^2 \tag{54}$$

$$= f(W_k) - \frac{\alpha}{2} \|\nabla f(W_k)\|^2 - (\frac{1}{2\alpha} - \frac{L}{2}) \|W_{k+1} - W_k\|^2$$
$$+ \frac{1}{2\alpha} \|W_{k+1} - W_k + \alpha \nabla f(W_k)\|^2$$
$$\leq f(W_k) - \frac{\alpha}{2} \|\nabla f(W_k)\|^2 + \frac{1}{2\alpha} \|W_{k+1} - W_k + \alpha \nabla f(W_k)\|^2$$

where the equality comes from

$$\langle \nabla f(W_k), W_{k+1} - W_k \rangle = - \frac{\alpha}{2} \|\nabla f(W_k)\|^2 - \frac{1}{2\alpha} \|W_{k+1} - W_k\|^2 \tag{55}$$
$$+ \frac{1}{2\alpha} \|W_{k+1} - W_k + \alpha \nabla f(W_k)\|^2.$$

The third term in the RHS of (54) can be bounded by

$$\frac{1}{2\alpha} \|W_{k+1} - W_k + \alpha \nabla f(W_k)\|^2 = \frac{\alpha}{2\tau^2} \||\nabla f(W_k)| \odot W_k\|^2 \tag{56}$$
$$\leq \frac{\alpha}{2\tau^2} \|\nabla f(W_k)\|^2 \|W_k\|_\infty^2.$$

Substituting (56) back into (54) and cooperating with Assumption 4 yield

$$f(W_{k+1}) \leq f(W_k) - \frac{\alpha}{2} \left( 1 - \frac{\|W_k\|_\infty^2}{\tau^2} \right) \|\nabla f(W_k)\|^2 \tag{57}$$
$$\leq f(W_k) - \frac{\alpha}{2} \left( 1 - \frac{W_{\max}^2}{\tau^2} \right) \|\nabla f(W_k)\|^2.$$

Rearranging and averaging (57) for $k$ from 0 to $K - 1$ deduce that

$$\frac{1}{K} \sum_{k=0}^{K-1} \|\nabla f(W_k)\|^2 \leq \frac{2(f(W_0) - f(W_k))}{\alpha K(1 - W_{\max}^2/\tau^2)}. \tag{58}$$

Noticing Assumption 2 claims that $f(W_k) \geq f^*$, we complete the proof. $\qquad\square$

## G Proof of Analog Stochastic Gradient Descent Convergence

### G.1 Proof of Theorem 2: Convergence of `Analog SGD`

This section provides the convergence guarantee of `Analog SGD` under non-convex assumption on asymmetric linear devices.

**Theorem 2** (Convergence of `Analog SGD`). *Under Assumption 1-4, if the learning rate is set as $\alpha = \sqrt{\frac{f(W_0) - f^*}{\sigma^2 L K}}$ and $K$ is sufficiently large such that $\alpha \leq \frac{1}{L}$, it holds that*

$$\frac{1}{K} \sum_{k=0}^{K-1} \mathbb{E}[\|\nabla f(W_k)\|^2] \leq O \left( \sqrt{\frac{(f(W_0) - f^*)\sigma^2 L}{K}} \frac{1}{1 - W_{\max}^2/\tau^2} \right) + \sigma^2 S_K \tag{9}$$

*where $S_K$ denotes the amplification factor given by $S_K := \frac{1}{K} \sum_{k=0}^{K} \frac{\|W_k\|_\infty^2/\tau^2}{1 - \|W_k\|_\infty^2/\tau^2} \leq \frac{W_{\max}^2/\tau^2}{1 - W_{\max}^2/\tau^2}$.*

*Proof of Theorem 2.* The $L$-smooth assumption (Assumption 1) implies that

$$\mathbb{E}_{\varepsilon_k}[f(W_{k+1})] \leq f(W_k) + \mathbb{E}_{\varepsilon_k}[\langle \nabla f(W_k), W_{k+1} - W_k \rangle] + \frac{L}{2} \mathbb{E}_{\varepsilon_k}[\|W_{k+1} - W_k\|^2] \tag{59}$$

$$\leq f(W_k) - \frac{\alpha}{2} \|\nabla f(W_k)\|^2 - (\frac{1}{2\alpha} - L) \mathbb{E}_{\varepsilon_k}[\|W_{k+1} - W_k + \alpha \varepsilon_k\|^2]$$

$$+ \alpha^2 L \mathbb{E}_{\varepsilon_k}[\|\varepsilon_k\|^2] + \frac{1}{2\alpha}\|W_{k+1} - W_k + \alpha(\nabla f(W_k) + \varepsilon_k)\|^2$$

where the second inequality comes from the assumption that noise has expectation 0 (Assumption 3)

$$\begin{aligned}
\mathbb{E}_{\varepsilon_k}[\langle \nabla f(W_k), W_{k+1} - W_k \rangle] &= \mathbb{E}_{\varepsilon_k}[\langle \nabla f(W_k), W_{k+1} - W_k + \alpha \varepsilon_k \rangle] \\
&= -\frac{\alpha}{2}\|\nabla f(W_k)\|^2 - \frac{1}{2\alpha}\mathbb{E}_{\varepsilon_k}[\|W_{k+1} - W_k + \alpha\varepsilon_k\|^2] \\
&\quad + \frac{1}{2\alpha}\mathbb{E}_{\varepsilon_k}[\|W_{k+1} - W_k + \alpha(\nabla f(W_k) + \varepsilon_k)\|^2]
\end{aligned}$$

and the following inequality

$$\frac{L}{2}\mathbb{E}_{\varepsilon_k}[\|W_{k+1} - W_k\|^2] \le L\mathbb{E}_{\varepsilon_k}[\|W_{k+1} - W_k + \alpha\varepsilon_k\|^2] + \alpha^2 L\mathbb{E}_{\varepsilon_k}[\|\varepsilon_k\|^2]. \quad (60)$$

With the learning rate $\alpha \le \frac{1}{2L}$ and bounded variance of noise (Assumption 3), (59) becomes

$$\mathbb{E}_{\varepsilon_k}[f(W_{k+1})] \le f(W_k) - \frac{\alpha}{2}\|\nabla f(W_k)\|^2 + \alpha^2 L\sigma^2 + \frac{1}{2\alpha}\|W_{k+1} - W_k + \alpha(\nabla f(W_k) + \varepsilon_k)\|^2. \tag{61}$$

Cooperated with Assumption 3, the last term in the RHS of (59) can be bounded by

$$\frac{1}{2\alpha}\mathbb{E}_{\varepsilon_k}[\|W_{k+1} - W_k + \alpha\nabla f(W_k) + \alpha\varepsilon_k\|^2] \tag{62}$$

$$= \frac{\alpha}{2\tau^2}\mathbb{E}_{\varepsilon_k}[\||\nabla f(W_k) + \varepsilon_k| \odot W_k\|^2]$$

$$\le \frac{\alpha}{2\tau^2}\mathbb{E}_{\varepsilon_k}[\|\nabla f(W_k) + \varepsilon_k\|^2]\|W_k\|_\infty^2$$

$$\le \frac{\alpha}{2\tau^2}\|\nabla f(W_k)\|^2\|W_k\|_\infty^2 + \frac{\alpha\sigma^2}{2\tau^2}\|W_k\|_\infty^2$$

where the last inequality comes from Assumption 3

$$\mathbb{E}_{\varepsilon_k}[\|\nabla f(W_k) + \varepsilon_k\|^2] = \|\nabla f(W_k)\|^2 + \mathbb{E}_{\varepsilon_k}[\|\varepsilon_k\|^2] \le \|\nabla f(W_k)\|^2 + \sigma^2. \quad (63)$$

Plugging (62) back into (59) yields

$$\frac{\alpha}{2}\left(1 - \frac{\|W_k\|_\infty^2}{\tau^2}\right)\|\nabla f(W_k)\|^2 \le \mathbb{E}_{\varepsilon_k}[f(W_k) - f(W_{k+1})] + \alpha^2 L\sigma^2 + \frac{\alpha\sigma^2\|W_k\|_\infty^2}{2\tau^2}. \quad (64)$$

Assumption 4 ensures that $1 - \|W_k\|_\infty^2/\tau^2 \ge 1 - W_{\max}/\tau^2 > 0$, which enables dividing both side of (64) by $1 - \|W_k\|_\infty^2/\tau^2$. Taking expectation over all $\varepsilon_k, \varepsilon_{k-1}, \cdots, \varepsilon_0$ and averaging for $k$ from 0 to $K-1$ deduce that

$$\frac{1}{K}\sum_{k=0}^{K}\mathbb{E}[\|\nabla f(W_k)\|^2] \tag{65}$$

$$\le \frac{2(f(W_0) - \mathbb{E}[f(W_{k+1})])}{\alpha K(1 - W_{\max}^2/\tau^2)} + \frac{2\alpha L\sigma^2}{1 - W_{\max}^2/\tau^2} + \frac{1}{K}\sum_{k=0}^{K}\frac{\sigma^2\|W_k\|_\infty^2/\tau^2}{1 - \|W_k\|_\infty^2/\tau^2}$$

$$\le \frac{2(f(W_0) - f^*)}{\alpha K(1 - W_{\max}^2/\tau^2)} + \frac{2\alpha L\sigma^2}{1 - W_{\max}^2/\tau^2} + \frac{1}{K}\sum_{k=0}^{K}\frac{\sigma^2\|W_k\|_\infty^2/\tau^2}{1 - \|W_k\|_\infty^2/\tau^2}$$

$$= 4\sqrt{\frac{(f(W_0) - f^*)\sigma^2 L}{K}}\frac{1}{1 - W_{\max}^2/\tau^2} + \sigma^2 S_K$$

where the second inequality uses Assumption 2 and the last equality chooses the learning rate as $\alpha = \sqrt{\frac{f(W_0)-f^*}{\sigma^2 LK}}$. The proof is completed. $\qquad \square$

## G.2 Proof of Theorem 3: Lower Bound of `Analog SGD`

This section provides the lower bound of Analog GD under non-convex assumption on noisy asymmetric linear devices.

**Theorem 3** (Lower bound of the error of `Analog SGD`). *There is an instance which satisfies Assumption 1-4 such that `Analog SGD` generates a sequence $\{W_k : k = 0, 1, \cdots, K\}$ which satisfies*

$$\frac{1}{K} \sum_{k=0}^{K-1} \mathbb{E}[\|\nabla f(W_k)\|^2] = \sigma^2 S_K + \Theta(\alpha) \overset{\alpha=\Theta\left(\frac{1}{\sqrt{K}}\right)}{=} \Omega\left(\sigma^2 S_K + \frac{1}{\sqrt{K}}\right). \tag{10}$$

The proof is completed based on the following example.

**(Example)** Consider an example where all the coordinates are identical, i.e. $W_k = w_k \mathbf{1}$ for some $w_k \in \mathbb{R}$ where $\mathbf{1} \in \mathbb{R}^D$ is the all-one vector. Define $W^* = w^* \mathbf{1}$ where $w^* \in \mathbb{R}$ is a constant scalar and a quadratic function $f(W) := \frac{L}{2}\|W - W^*\|^2$ whose minimum is $W^*$. Initialize the weight on $W_0 = W^*$ and choose the learning rate $\alpha \leq \min\{\frac{1}{2L}, \frac{1}{\mu+6\sigma/(\tau\sqrt{D})}\}$. Furthermore, consider the noise defined as follows,

$$\varepsilon_k = \xi_k \mathbf{1} \tag{66}$$

where random variable $\xi_k \in \mathbb{R}$ is sampled by

$$\xi_k = \begin{cases} \xi_k^+ := \frac{\sigma}{\sqrt{D}}\sqrt{\frac{1-p_k}{p_k}}, & \text{w.p. } p_k, \\ \xi_k^- := -\frac{\sigma}{\sqrt{D}}\sqrt{\frac{p_k}{1-p_k}}, & \text{w.p. } 1-p_k, \end{cases} \quad \text{with } p_k = \frac{1}{2}\left(1 - \frac{w_k}{\tau}\right). \tag{67}$$

As a reminder, it is always valid that $|w_k| = \|W_k\|_\infty \leq \tau$ (c.f. Theorem 5) and $0 \leq p_k \leq 1$. Therefore, the noise distribution is well-defined. Furthermore, without loss of generality[3], we assume $|w^*| \leq \frac{\tau}{4}$ and $\sigma \leq \frac{\tau L\sqrt{D}}{4\sqrt{3}}$.

Since all the coordinates are identical, `Analog SGD` can be regarded to train a scalar weight at each coordinate. Furthermore, with the definition

$$f(w; \xi_k) := \frac{L}{2}(w - w^* + \frac{\xi_k}{L})^2, \quad \text{whose minimum is } w_{\xi_k}^* = w^* - \frac{\xi_k}{L}, \tag{68}$$

the noise $\xi_k$ satisfies Assumption 7. Therefore, even though $w_{\xi_k}^*$ are time varying, we can regard $w_k$ as the initial point and only consider $w_{k+1}$. By this way, the conditions of Theorem 6 and Theorem 8 are met, and their claim for $w_{k+1}$ is valid.

To prove the lower bound, we next introduce several necessary lemmas to facilitate the proof of Theorem 3. The first one is used to verify Assumption 4.

**Lemma 2.** *Suppose $|w^*| \leq \frac{\tau}{4}$ and $\sigma \leq \frac{\tau L\sqrt{D}}{4\sqrt{3}}$. Define $W_{\max} := \frac{\tau}{2}$. The sequence $\{W_k : k \in \mathbb{N}\}$ generated by `Analog SGD` (3) on the example above satisfies*

$$\|W_k\|_\infty \leq W_{\max}. \tag{69}$$

*Proof of Lemma 2.* The statement can be proved by induction. When $k = 0$,

$$\|W_0\|_\infty = |w^*| \leq \frac{\tau}{4} < W_{\max}. \tag{70}$$

To prove (69) for $k + 1$, Theorem 8 asserts that

$$\min\left\{W_0, \inf_{[\varepsilon]_i}\{w_{[\varepsilon]_i}^*\}\right\} \leq [W_{k+1}]_i \leq \max\left\{W_0, \sup_{[\varepsilon]_i}\{w_{[\varepsilon]_i}^*\}\right\} \tag{71}$$

or equivalently

$$w^* - \frac{\sigma}{L\sqrt{D}}\sqrt{\frac{1-p_k}{p_k}} \leq [W_{k+1}]_i \leq w^* + \frac{\sigma}{L\sqrt{D}}\sqrt{\frac{p_k}{1-p_k}}. \tag{72}$$

---

[3]These requirements are necessitated by Lemma 2 and can be relaxed to $|w^*| \leq c_*\tau$ for any constant $c_* < 1$. In that situation, Lemma 2 remains valid, although $W_{\max}$ differs.

According to the triangle inequality, $\|W_{k+1}\|_\infty$ is still bounded by

$$\|W_{k+1}\|_\infty = \max_i\{|[W_{k+1}]_i|\} \leq |w^*| + \frac{\sigma}{L\sqrt{D}} \max\left\{\sqrt{\frac{1-p_k}{p_k}}, \sqrt{\frac{p_k}{1-p_k}}\right\} \tag{73}$$

$$= |w^*| + \frac{\sigma}{L\sqrt{D}} \sqrt{\frac{1 + \|W_k\|_\infty/\tau}{1 - \|W_k\|_\infty/\tau}}$$

$$\leq |w^*| + \frac{\sigma}{L\sqrt{D}} \sqrt{\frac{1 + W_{\max}/\tau}{1 - W_{\max}/\tau}}$$

$$\leq \frac{\tau}{4} + \frac{\tau}{4\sqrt{3}} \sqrt{\frac{1 + (2\tau)/\tau}{1 - (2\tau)/\tau}} = \frac{\tau}{2}$$

$$= W_{\max}.$$

Therefore, (69) is confirmed for $k+1$ and the proof of Lemma 2 is completed. $\qquad\square$

Noise $\xi_k$ defined by (67) is time-varying since $W_k$ is changing. The following lemma provides an upper bound for the variation of noise with respect to $W_k$.

**Lemma 3.** *Suppose $\|W_k\|_\infty \leq W_{\max} \leq \frac{\tau}{2}$. The following statements are always valid*

$$|\xi_{k+1}^+ - \xi_k^+| \leq \frac{3\sigma}{\tau\sqrt{D}}|w_{k+1} - w_k|, \tag{74a}$$

$$|\xi_{k+1}^- - \xi_k^-| \leq \frac{3\sigma}{\tau\sqrt{D}}|w_{k+1} - w_k|. \tag{74b}$$

*Proof of Lemma 3.* Define the functions

$$g^+(x) := \sqrt{\frac{1+x}{1-x}}, \quad g^-(x) := \sqrt{\frac{1-x}{1+x}}, \tag{75}$$

with which $\xi_k^+$ and $\xi_k^-$ can be written as

$$\xi_k^+ = \frac{\sigma}{\sqrt{D}} g^+\left(\frac{w_k}{\tau}\right), \quad \xi_k^- = \frac{\sigma}{\sqrt{D}} g^-\left(\frac{w_k}{\tau}\right). \tag{76}$$

The derivatives of $g^+(x)$ and $g^-(x)$ is

$$\nabla g^+(x) = \frac{1}{2}\left(\frac{1}{\sqrt{(1+x)(1-x)}} + \sqrt{\frac{1+x}{(1-x)^3}}\right), \tag{77}$$

$$\nabla g^-(x) = -\frac{1}{2}\left(\frac{1}{\sqrt{(1+x)(1-x)}} + \sqrt{\frac{1-x}{(1+x)^3}}\right), \tag{78}$$

whose norms are upper bounded by

$$|\nabla g^+(x)| \leq \frac{1}{2}\left(\frac{2}{\sqrt{3}} + 2\sqrt{3}\right) \leq 3, \tag{79}$$

$$|\nabla g^-(x)| \leq \frac{1}{2}\left(\frac{2}{\sqrt{3}} + 2\sqrt{3}\right) \leq 3. \tag{80}$$

Therefore, both $g^+(x)$ and $g^-(x)$ are Lipschitz continuous and hence

$$|\xi_{k+1}^+ - \xi_k^+| = \frac{\sigma}{\sqrt{D}}\left|g^+\left(\frac{w_{k+1}}{\tau}\right) - g^+\left(\frac{w_k}{\tau}\right)\right| \leq \frac{3\sigma}{\tau\sqrt{D}}|w_{k+1} - w_k|, \tag{81}$$

$$|\xi_{k+1}^- - \xi_k^-| = \frac{\sigma}{\sqrt{D}}\left|g^-\left(\frac{w_{k+1}}{\tau}\right) - g^-\left(\frac{w_k}{\tau}\right)\right| \leq \frac{3\sigma}{\tau\sqrt{D}}|w_{k+1} - w_k|. \tag{82}$$

The proof of Lemma 3 is then completed. $\qquad\square$

In the proof of Theorem 3, the most tricky part in the recursion (3) is the gradient wrapped by absolute value $|\nabla f(W_k) + \varepsilon_k|$. Fortunately, the expectation $\mathbb{E}[|\nabla f(W_k) + \varepsilon_k|]$ can be expressed explicitly in the example above, which is demonstrated by the following lemma.

**Lemma 4.** *Suppose the learning rate is chosen such that $\alpha \leq \min\{\frac{1}{2L}, \frac{1}{\mu + 6\sigma/(\tau\sqrt{D})}\}$ and (74) are valid, it holds that*

$$\mathbb{E}[|\nabla f(W_k) + \varepsilon_k|] = \frac{2\sigma}{\sqrt{D}}\sqrt{p_k(1 - p_k)} \cdot \mathbf{1} + (2p_k - 1)\nabla f(W_k). \tag{83}$$

*Proof of Lemma 4.* The proof of Lemma 4 closely relies on the following inequality

$$w^* - \frac{\xi_k^+}{L} \leq [W_k]_i \leq w^* + \frac{\xi_k^-}{L} \tag{84}$$

or equivalently, the gradient $[\nabla f(W_k)]_i = L([W_k]_i - w^*)$ satisfies

$$\xi_k^+ \leq [\nabla f(W_k)]_i \leq \xi_k^-, \quad \forall i \in \mathcal{I}. \tag{85}$$

By (85), we claim the signs of $[\nabla f(W_k)]_i + \xi_k^+$ and $[\nabla f(W_k)]_i + \xi_k^-$ never change during the training and thus the absolute value can be removed

$$|\nabla f(W_k) + \xi_k^+ \mathbf{1}| = \xi_k^+ \mathbf{1} + \nabla f(W_k), \tag{86}$$
$$|\nabla f(W_k) - \xi_k^- \mathbf{1}| = \xi_k^- \mathbf{1} - \nabla f(W_k). \tag{87}$$

Accordingly, $\mathbb{E}[|\nabla f(W_k) + \varepsilon_k|]$ can be written as

$$\begin{aligned}
&\mathbb{E}[|\nabla f(W_k) + \varepsilon_k|] \\
&= p_k\left(\xi_k^+ \mathbf{1} + \nabla f(W_k)\right) + (1 - p_k)\left(\xi_k^- \mathbf{1} - \nabla f(W_k)\right) \\
&= p_k\left(\frac{\sigma}{\sqrt{D}}\sqrt{\frac{1 - p_k}{p_k}} \cdot \mathbf{1} + \nabla f(W_k)\right) + (1 - p_k)\left(\frac{\sigma}{\sqrt{D}}\sqrt{\frac{p_k}{1 - p_k}} \cdot \mathbf{1} - \nabla f(W_k)\right) \\
&= \frac{2\sigma}{\sqrt{D}}\sqrt{p_k(1 - p_k)} \cdot \mathbf{1} + (2p_k - 1)\nabla f(W_k)
\end{aligned} \tag{88}$$

which is the result. Therefore, the rest of proof shows (84) is valid.

**Verification of** (84). The statement can be proved by induction. When $k = 0$, (84) holds naturally since $W_0 = W^*$.

To prove (84) for $k + 1$, we consider $\varepsilon_k = \xi_k^+ \mathbf{1}$ and $\varepsilon_k = \xi_k^- \mathbf{1}$ separately, and the conclusion holds for both cases.

**Case 1:** $\varepsilon_k = \xi_k^+ \mathbf{1} = \frac{\sigma}{\sqrt{D}}\sqrt{\frac{1 - p_k}{p_k}} \cdot \mathbf{1}$. From the induction assumption, it is valid that

$$[W_k]_i \geq w^* - \frac{\sigma}{L\sqrt{D}}\sqrt{\frac{1 - p_k}{p_k}} = w^*_{[\varepsilon]_i}. \tag{89}$$

Therefore, Theorem 6 asserts that $[W_{k+1}]_i \leq [W_k]_i$ and hence we have $p_{k+1} \geq p_k$, $\xi_{k+1}^+ \leq \xi_k^+$ and $\xi_{k+1}^- \leq \xi_k^-$. Consequently, the second inequality of (84) can be reached by Theorem 8

$$[W_{k+1}]_i \leq w^* - \frac{\xi_k^-}{L} \leq w^* - \frac{\xi_{k+1}^-}{L}. \tag{90}$$

To obtain the other inequality, notice that

$$\begin{aligned}
w^* - \frac{\xi_{k+1}^+}{L} &= [W_{k+1}]_i + \underbrace{(w^* - \frac{\xi_{k+1}^+}{L} - [W_{k+1}]_i)}_{\leq 0} + \frac{1}{L}\underbrace{(\xi_k^+ - \xi_{k+1}^+)}_{\geq 0} \\
&\overset{(a)}{=} [W_{k+1}]_i + (1 - \alpha L(1 - \frac{[W_k]_i}{\tau}))(w^* - \frac{\xi_{k+1}^+}{L} - [W_k]_i) + \frac{1}{L}(\xi_k^+ - \xi_{k+1}^+) \\
&\overset{(b)}{\leq} [W_{k+1}]_i + (1 - \alpha L(1 - \frac{[W_k]_i}{\tau}))(w^* - \frac{\xi_{k+1}^+}{L} - [W_k]_i) + \frac{6\alpha\sigma}{\tau\sqrt{D}}([W_k]_i - (w^* - \frac{\xi_{k+1}^+}{L}))
\end{aligned} \tag{91}$$

$$= [W_{k+1}]_i - (1 - \alpha L(1 - \frac{[W_k]_i}{\tau}) - \frac{6\alpha\sigma}{\tau\sqrt{D}})([W_k]_i - (w^* - \frac{\xi_{k+1}^+}{L}))$$

where $(a)$ comes from Theorem 6, (b) comes from Lemma 3 and inequality (37).

$$\xi_k^+ - \xi_{k+1}^+ = |\xi_k^+ - \xi_{k+1}^+| \le \frac{3}{\tau\sqrt{D}}|[W_{k+1}]_i - [W_k]_i| \le \frac{6\alpha\sigma}{\tau\sqrt{D}}|[W_k]_i - (w^* - \frac{\xi_{k+1}^+}{L})|. \quad (92)$$

The choice of learning rate implies that

$$1 - \alpha L(1 - \frac{[W_k]_i}{\tau}) - \frac{6\alpha\sigma}{\tau\sqrt{D}} \ge 1 - \alpha(L + \frac{6\sigma}{\tau\sqrt{D}}) \ge 0 \quad (93)$$

from which and (91) we reach that

$$w^* - \frac{\xi_{k+1}^+}{L} \le [W_{k+1}]_i. \quad (94)$$

Combining (94) and (90) reaches (84) in case 1.

**Case 2:** $\varepsilon_k = \xi_k^- \mathbf{1} = -\frac{\sigma}{\sqrt{D}}\sqrt{\frac{p_k}{1-p_k}} \cdot \mathbf{1}$. Noticing the permutation invariant between $p_k$ and $1 - p_k$ in the noise definition (67), we find that the proof for case 2 is similar to that of case 1. Therefore, the proof of case 2 is omitted here.

In conclusion, (84) still holds for $k + 1$ and (84) is verified. Now the proof of Lemma 4 is completed.
$\square$

After providing the necessary lemmas, we begin to prove Theorem 3.

*Proof of Theorem 3.* Consider the example constructed above. Before deriving the lower bound, we demonstrate Assumption 1–4 hold. It is obvious that $\nabla f(W) = L(W - W^*)$ satisfies Assumption 1 and $f(W) \ge f^* := 0$ satisfies Assumption 2. In addition, Assumption 3 could be verified by noticing (66) impplies $\mathbb{E}_{\varepsilon_k}[\varepsilon_k] = 0$ and $\mathbb{E}_{\varepsilon_k}[\|\varepsilon_k\|^2] \le \sigma^2$. Assumption 4 is verified by Lemma 2.

Now we can derive the lower bound. Utilizing Lemma 4 and manipulating the recursion (3), we have the following result

$$\mathbb{E}_{\varepsilon_k}[W_{k+1} - W^*] \quad (95)$$
$$= \mathbb{E}_{\varepsilon_k}[W_k - \alpha\nabla f(W_k) - \alpha\varepsilon_k - \frac{\alpha}{\tau}|\nabla f(W_k) + \varepsilon_k| \odot W_k - W^*]$$
$$= (1 - \alpha L)(W_k - W^*) - \frac{\alpha}{\tau}\mathbb{E}_{\varepsilon_k}[|\nabla f(W_k) + \varepsilon_k|] \odot W_k$$
$$= (1 - \alpha L)(W_k - W^*) - \frac{2\alpha\sigma}{\tau\sqrt{D}}\sqrt{p_k(1 - p_k)} \cdot W_k - \frac{\alpha}{\tau}(2p_k - 1)\nabla f(W_k) \odot W_k$$

Multiplying the both size of (95) by $L$ and plugging in the equation $\nabla f(W_k) = L(W_k - W^*)$ yield

$$\mathbb{E}_{\varepsilon_k}[\nabla f(W_{k+1})] \quad (96)$$
$$= (1 - \alpha L)\nabla f(W_k) - \frac{2\alpha L\sigma}{\tau\sqrt{D}}\sqrt{p_k(1 - p_k)} \cdot W_k - \frac{\alpha L}{\tau}(2p_k - 1)\nabla f(W_k) \odot W_k$$
$$= \left(1 - \alpha L - \alpha L(2p_k - 1)\frac{w_k}{\tau}\right)\nabla f(W_k) - \frac{2\alpha L\sigma}{\tau\sqrt{D}}\sqrt{p_k(1 - p_k)} \cdot W_k.$$

where the last equality uses $W_k = w_k\mathbf{1}$. Recall the probability $p_k = \frac{1}{2}(1 - \frac{w_k}{\tau})$, we have

$$1 - p_k = \frac{1}{2}\left(1 + \frac{w_k}{\tau}\right), \quad \sqrt{p_k(1 - p_k)} = \frac{1}{2}\sqrt{1 - \frac{w_k^2}{\tau^2}}, \quad 2p_k - 1 = -\frac{w_k}{\tau}. \quad (97)$$

Substitute them back into (96)

$$\mathbb{E}_{\varepsilon_k}[\nabla f(W_{k+1})] = \left(1 - \alpha L + \alpha L\frac{|w_k|^2}{\tau^2}\right)\nabla f(W_k) - \frac{\alpha L\sigma}{\tau\sqrt{D}}\sqrt{1 - \frac{|w_k|^2}{\tau^2}} \cdot W_k \quad (98)$$

$$= \left(1 - \alpha L + \alpha L \frac{\|W_k\|_\infty^2}{\tau^2}\right) \nabla f(W_k) - \frac{\alpha L \sigma}{\tau \sqrt{D}} \sqrt{1 - \frac{\|W_k\|_\infty^2}{\tau^2}} \cdot W_k$$

$$= \nabla f(W_k) - \alpha L \left(1 - \frac{\|W_k\|_\infty^2}{\tau^2}\right) \nabla f(W_k) - \frac{\alpha L \sigma}{\tau \sqrt{D}} \sqrt{1 - \frac{\|W_k\|_\infty^2}{\tau^2}} \cdot W_k$$

where the first equality utilizes $|w_k|^2 = \|w_k \mathbf{1}\|_\infty^2 = \|W_k\|_\infty^2$. Reorganizing (98), we obtain

$$\nabla f(W_k) = -\frac{\sigma}{\sqrt{1 - \|W_k\|_\infty^2/\tau^2}} \frac{W_k/\tau}{\sqrt{D}} + \frac{\nabla f(W_k) - \mathbb{E}_{\varepsilon_k}[\nabla f(W_{k+1})]}{1 - \|W_k\|_\infty^2/\tau^2}.$$

It is worth noticing that the second term in the RHS of (99) is in the order of $O(\alpha)$

$$\left\| \frac{\nabla f(W_k) - \mathbb{E}_{\varepsilon_k}[\nabla f(W_{k+1})]}{1 - \|W_k\|_\infty^2/\tau^2} \right\| = L \left\| \frac{\mathbb{E}_{\varepsilon_k}[W_k - W_{k+1}]}{1 - \|W_k\|_\infty^2/\tau^2} \right\|$$

$$\overset{(a)}{\leq} \frac{\alpha L \|\mathbb{E}_{\varepsilon_k}[\nabla f(W_k) + \varepsilon_k]\|}{(1 - \|W_k\|_\infty/\tau)(1 - \|W_k\|_\infty^2/\tau^2)}$$

$$= \frac{\alpha L \|\nabla f(W_k)\|}{(1 - \|W_k\|_\infty/\tau)(1 - \|W_k\|_\infty^2/\tau^2)}$$

$$\overset{(b)}{\leq} \frac{\alpha L^2 (W_{\max} + W^*)D}{(1 - W_{\max}/\tau)(1 - W_{\max}^2/\tau^2)}$$

$$= O(\alpha)$$

where $(a)$ is Lemma 1 and $(b)$ comes from $\|\nabla f(W_k)\| = L\|W_k - W^*\| \leq L(\|W_k\| + \|W^*\|) \leq L\sqrt{D}(\|W_k\|_\infty + |w^*|)$. Therefore, the gradient can be rewritten as

$$\nabla f(W_k) = -\frac{\sigma}{\sqrt{1 - \|W_k\|_\infty^2/\tau^2}} \frac{W_k/\tau}{\sqrt{D}} + O(\alpha). \tag{99}$$

Taking the square norm of the gradient and averaging for $k$ from 0 to $K-1$ we obtain

$$\frac{1}{K} \sum_{k=0}^{K} \|\nabla f(W_k)\|^2 = \sigma^2 \frac{1}{K} \sum_{k=0}^{K} \frac{\|W_k\|_\infty^2/\tau^2}{1 - \|W_k\|_\infty^2/\tau^2} = \sigma^2 S_K + O(\alpha) \tag{100}$$

where the following equality is applied

$$\left\| \frac{W_k/\tau}{\sqrt{D}} \right\|^2 = \left\| \frac{\|W_k\|_\infty \mathbf{1}}{\sqrt{D}} \right\|^2 = \|W_k\|_\infty^2/\tau^2. \tag{101}$$

The proof of Theorem 3 is thus completed. $\qquad\square$

# H  Proof of `Tiki-Taka` Convergence

This section provides the convergence guarantee of the `Tiki-Taka` under the non-convex assumption.

**Theorem 4** (Convergence of `Tiki-Taka`). *Suppose Assumption 1-5 hold and the learning rate is set as $\alpha = O(1/\sqrt{\sigma^2 K})$, $\beta = 8\alpha L$. It holds for $Tiki\text{-}Taka$ that the expected infinity norm $P_k$ is upper bounded by $\mathbb{E}[\|P_{k+1}\|_\infty^2] \leq P_{\max}^2 := \frac{41L^2\tau^4 D}{c^2\sigma^2}$. Furthermore, if $\sigma^2$ and $D$ are sufficiently large so that $33P_{\max}^2/\tau^2 < 1$ it is valid that*

$$\frac{1}{K} \sum_{k=0}^{K-1} \mathbb{E}[\|\nabla f(W_k)\|^2] \leq O\left( \sqrt{\frac{(f(W_0) - f^*)\sigma^2 L}{K}} \frac{1}{1 - 33P_{\max}^2/\tau^2} \right). \tag{14}$$

## H.1  Proof of Tracking Lemma

Before the proving Theorem 4, we first introduce two useful lemmas.

**Lemma 5** (Tracking Lemma). *Under Assumption 1 and 3, it holds that*

$$\mathbb{E}_{\varepsilon_{k+1}}[\|P_{k+2} - \frac{\tau\sqrt{D}}{\sigma}\nabla f(W_{k+1})\|^2] \tag{102}$$

$$\leq \left(1 - \frac{\beta\sigma}{\tau\sqrt{D}}\right)\|P_{k+1} - \frac{\tau\sqrt{D}}{\sigma}\nabla f(W_k)\|^2 + \frac{4L^2\tau^3 D^{3/2}}{\beta\sigma^3}\|W_{k+1} - W_k\|^2 + \frac{4\beta\sigma}{\tau\sqrt{D}}\|P_{k+1}\|^2$$

$$+ \frac{4\beta\sqrt{D}}{\tau\sigma}\|P_{k+1}\|_\infty^2\|\nabla f(W_k)\|^2 + 2\beta^2\sigma^2.$$

To make $P_{k+2}$ sufficiently close to the true gradient, $\|P_{k+1}\|^2$ should be small.

*Proof of Lemma 5.* According to the update rule of $P_k$, we have

$$\mathbb{E}_{\varepsilon_{k+1}}[\|P_{k+2} - \frac{\tau\sqrt{D}}{\sigma}\nabla f(W_{k+1})\|^2] \tag{103}$$

$$= \mathbb{E}_{\varepsilon_{k+1}}[\|P_{k+1} + \beta\nabla f(W_{k+1}) + \beta\varepsilon_{k+1} - \frac{\beta}{\tau}|\nabla f(W_{k+1}) + \varepsilon_{k+1}| \odot P_{k+1} - \frac{\tau\sqrt{D}}{\sigma}\nabla f(W_{k+1})\|^2]$$

$$= \mathbb{E}_{\varepsilon_{k+1}}[\|(1 - \frac{\beta\sigma}{\tau\sqrt{D}})(P_{k+1} - \frac{\tau\sqrt{D}}{\sigma}\nabla f(W_{k+1})) + \frac{\beta\sigma}{\tau\sqrt{D}}P_{k+1}$$

$$- \frac{\beta}{\tau}|\nabla f(W_{k+1}) + \varepsilon_{k+1}| \odot P_{k+1} + \beta\varepsilon_{k+1}\|^2]$$

$$\leq \frac{1}{1-u}\mathbb{E}_{\varepsilon_{k+1}}[\|(1 - \frac{\beta\sigma}{\tau\sqrt{D}})(P_{k+1} - \frac{\tau\sqrt{D}}{\sigma}\nabla f(W_{k+1})) + \beta\varepsilon_{k+1}\|^2]$$

$$+ \frac{\beta^2}{u\tau^2}\mathbb{E}_{\varepsilon_{k+1}}[\|(\sigma\mathbf{1}/\sqrt{D} - |\nabla f(W_{k+1}) + \varepsilon_{k+1}|) \odot P_{k+1}\|^2].$$

With Assumption 3, the first term of (103) can be bounded via the same technique as (63), that is

$$\frac{1}{1-u}\mathbb{E}_{\varepsilon_{k+1}}[\|(1 - \frac{\beta\sigma}{\tau\sqrt{D}})(P_{k+1} - \frac{\tau\sqrt{D}}{\sigma}\nabla f(W_{k+1})) + \beta\varepsilon_{k+1}\|^2] \tag{104}$$

$$\leq \frac{(1 - \frac{\beta\sigma}{\tau\sqrt{D}})^2}{1-u}\|P_{k+1} - \frac{\tau\sqrt{D}}{\sigma}\nabla f(W_{k+1})\|^2 + \frac{\beta^2}{1-u}\sigma^2$$

$$\leq \left(1 - \frac{2\beta\sigma}{\tau\sqrt{D}} + u\right)\|P_{k+1} - \frac{\tau\sqrt{D}}{\sigma}\nabla f(W_{k+1})\|^2 + \frac{\beta^2}{1-u}\sigma^2$$

where the last inequality holds because

$$(1 - \frac{\beta\sigma}{\tau\sqrt{D}})^2 \leq (1-u)\left(1 - \frac{2\beta\sigma}{\tau\sqrt{D}} + u\right). \tag{105}$$

The third term in the RHS of (103) can be manipulated by the variance decomposition and consequently can be bounded by

$$\frac{\beta^2}{u\tau^2}\mathbb{E}_{\varepsilon_{k+1}}[\|(\sigma\mathbf{1}/\sqrt{D} - |\nabla f(W_{k+1}) + \varepsilon_{k+1}|) \odot P_{k+1}\|^2] \tag{106}$$

$$= \frac{\beta^2}{u\tau^2}\mathbb{E}_{\varepsilon_{k+1}}\left[\sum_{i\in\mathcal{I}}(\sigma/\sqrt{D} - |[\nabla f(W_{k+1}) + \varepsilon_{k+1}]_i|)^2[P_{k+1}]_i^2\right]$$

$$= \frac{\beta^2}{u\tau^2}\mathbb{E}_{\varepsilon_{k+1}}\left[\sum_{i\in\mathcal{I}}(\sigma^2/D - 2\sigma|[\nabla f(W_{k+1}) + \varepsilon_{k+1}]_i| + [\nabla f(W_{k+1}) + \varepsilon_{k+1}]_i^2)[P_{k+1}]_i^2\right]$$

$$= \frac{\beta^2}{u\tau^2}\mathbb{E}_{\varepsilon_{k+1}}\left[\sum_{i\in\mathcal{I}}(\sigma^2/D - 2\sigma|[\nabla f(W_{k+1}) + \varepsilon_{k+1}]_i|)[P_{k+1}]_i^2\right]$$

$$+ \frac{\beta^2}{u\tau^2}\mathbb{E}_{\varepsilon_{k+1}}\left[\sum_{i\in\mathcal{I}}([\nabla f(W_{k+1}) + \varepsilon_{k+1}]_i^2)[P_{k+1}]_i^2\right]$$

$$\leq \frac{\beta^2}{u\tau^2}(\sigma^2/D - 2c\sigma^2 + \sigma^2/D)\|P_{k+1}\|^2 + \frac{\beta^2}{u\tau^2}\|P_{k+1}\|_\infty^2\|\nabla f(W_{k+1})\|^2$$

$$\leq \frac{2\beta^2\sigma^2}{u\tau^2 D}\|P_{k+1}\|^2 + \frac{\beta^2}{u\tau^2}\|P_{k+1}\|_\infty^2\|\nabla f(W_{k+1})\|^2.$$

Combining (104) and (106) with (103) provides that

$$\mathbb{E}_{\varepsilon_{k+1}}[\|P_{k+2} - \frac{\tau\sqrt{D}}{\sigma}\nabla f(W_{k+1})\|^2] \tag{107}$$

$$\leq \left(1 - \frac{2\beta\sigma}{\tau\sqrt{D}} + u\right)\|P_{k+1} - \frac{\tau\sqrt{D}}{\sigma}\nabla f(W_{k+1})\|^2 + \frac{\beta^2}{1-u}\sigma^2 + \frac{2\beta^2\sigma^2}{u\tau^2 D}\|P_{k+1}\|^2$$

$$+ \frac{\beta^2}{u\tau^2}\|P_{k+1}\|_\infty^2\|\nabla f(W_{k+1})\|^2$$

$$\overset{(a)}{\leq} \left(1 - \frac{3\beta\sigma}{2\tau\sqrt{D}}\right)\|P_{k+1} - \frac{\tau\sqrt{D}}{\sigma}\nabla f(W_{k+1})\|^2 + 2\beta^2\sigma^2 + \frac{4\beta\sigma}{\tau\sqrt{D}}\|P_{k+1}\|^2$$

$$+ \frac{2\beta\sqrt{D}}{\tau\sigma}\|P_{k+1}\|_\infty^2\|\nabla f(W_{k+1})\|^2$$

$$\leq \left(1 - \frac{3\beta\sigma}{2\tau\sqrt{D}}\right)\left(1 + \frac{\beta\sigma}{2\tau\sqrt{D}}\right)\|P_{k+1} - \frac{\tau\sqrt{D}}{\sigma}\nabla f(W_k)\|^2$$

$$+ \left(1 - \frac{3\beta\sigma}{2\tau\sqrt{D}}\right)\left(1 + \frac{2\tau\sqrt{D}}{\beta\sigma}\right)\frac{\tau^2 D}{\sigma^2}\|\nabla f(W_{k+1}) - \nabla f(W_k)\|^2 + 2\beta^2\sigma^2$$

$$+ \frac{4\beta\sigma}{\tau\sqrt{D}}\|P_{k+1}\|^2 + \frac{2\beta\sqrt{D}}{\tau\sigma}\|P_{k+1}\|_\infty^2\|\nabla f(W_{k+1})\|^2$$

$$\overset{(b)}{\leq} \left(1 - \frac{\beta\sigma}{\tau\sqrt{D}}\right)\|P_{k+1} - \frac{\tau\sqrt{D}}{\sigma}\nabla f(W_k)\|^2 + \frac{2L^2\tau^3 D^{3/2}}{\beta\sigma^3}\|W_{k+1} - W_k\|^2 + \frac{4\beta\sigma}{\tau\sqrt{D}}\|P_{k+1}\|^2$$

$$+ \frac{2\beta\sqrt{D}}{\tau\sigma}\|P_{k+1}\|_\infty^2\|\nabla f(W_{k+1})\|^2 + 2\beta^2\sigma^2,$$

$$\overset{(c)}{\leq} \left(1 - \frac{\beta\sigma}{\tau\sqrt{D}}\right)\|P_{k+1} - \frac{\tau\sqrt{D}}{\sigma}\nabla f(W_k)\|^2 + \frac{4L^2\tau^3 D^{3/2}}{\beta\sigma^3}\|W_{k+1} - W_k\|^2 + \frac{4\beta\sigma}{\tau\sqrt{D}}\|P_{k+1}\|^2$$

$$+ \frac{4\beta\sqrt{D}}{\tau\sigma}\|P_{k+1}\|_\infty^2\|\nabla f(W_k)\|^2 + 2\beta^2\sigma^2.$$

In the inequality above, $(a)$ sets $u = \beta/2$; $(b)$ holds because

$$\left(1 - \frac{3\beta\sigma}{2\tau\sqrt{D}}\right)\left(1 + \frac{\beta\sigma}{2\tau\sqrt{D}}\right) = 1 - \frac{\beta\sigma}{\tau\sqrt{D}} - \frac{3\beta\sigma}{2\tau\sqrt{D}}\frac{\beta\sigma}{2\tau\sqrt{D}} \leq 1 - \frac{\beta\sigma}{\tau\sqrt{D}} \tag{108}$$

and

$$\left(1 - \frac{3\beta\sigma}{2\tau\sqrt{D}}\right)\left(1 + \frac{2\tau\sqrt{D}}{\beta\sigma}\right) = \frac{2\tau\sqrt{D}}{\beta\sigma}\left(1 - \frac{3\beta\sigma}{2\tau\sqrt{D}}\right)\left(1 + \frac{\beta\sigma}{2\tau\sqrt{D}}\right) \leq \frac{2\tau\sqrt{D}}{\beta\sigma}; \tag{109}$$

and $(c)$ comes from

$$\frac{2\beta\sqrt{D}}{\tau\sigma}\|P_{k+1}\|_\infty^2\|\nabla f(W_{k+1})\|^2 \tag{110}$$

$$\leq \frac{4\beta\sqrt{D}}{\tau\sigma}\|P_{k+1}\|_\infty^2\|\nabla f(W_k)\|^2 + \frac{4\beta\sqrt{D}}{\tau\sigma}\|P_{k+1}\|_\infty^2\|\nabla f(W_{k+1}) - \nabla f(W_k)\|^2$$

$$\leq \frac{4\beta\sqrt{D}}{\tau\sigma}\|P_{k+1}\|_\infty^2\|\nabla f(W_k)\|^2 + \frac{4\beta L^2\tau\sqrt{D}}{\sigma}\|W_{k+1} - W_k\|^2$$

$$\leq \frac{4\beta\sqrt{D}}{\tau\sigma}\|P_{k+1}\|_\infty^2\|\nabla f(W_k)\|^2 + \frac{2L^2\tau^3 D^{3/2}}{\beta\sigma^3}\|W_{k+1} - W_k\|^2.$$

where the last inequality holds because $\beta \leq \frac{2\tau^2 D}{\beta\sigma^2}$. Now we get the claim. $\qquad\square$

## H.2 Proof of Weight Decay Lemma

**Lemma 6** (Weight Decay Lemma). *Suppose Assumption 1, 3 and 5 hold. If $\beta$ satisfies*

$$\beta \leq \min\left\{\frac{c\tau}{2(4L^2\tau^2+\sigma^2)}, \frac{2}{3c\sigma}\right\}, \tag{111}$$

*it holds that*

$$\mathbb{E}_{\varepsilon_{k+1}}[\|P_{k+2}\|^2] \leq (1-\frac{\beta c\sigma}{2\tau})\|P_{k+1}\|^2 + \frac{4\beta\tau}{c\sigma}\|\nabla f(W_k)\|^2 + \frac{4L^2\beta\tau}{c\sigma}\|W_{k+1}-W_k\|^2 + 3\beta^2\sigma^2.$$

*Proof of Lemma 6.* The proof begins from the manipulation of the expected square norm of $P_{k+2}$

$$\mathbb{E}_{\varepsilon_{k+1}}[\|P_{k+2}\|^2] \tag{112}$$

$$= \mathbb{E}_{\varepsilon_{k+1}}[\|P_{k+1} + \beta(\nabla f(W_{k+1}) + \varepsilon_{k+1}) - \frac{\beta}{\tau}|\nabla f(W_{k+1}) + \varepsilon_{k+1}| \odot P_{k+1}\|^2]$$

$$= \mathbb{E}_{\varepsilon_{k+1}}[\|(\mathbf{1} - \frac{\beta}{\tau}|\nabla f(W_{k+1}) + \varepsilon_{k+1}|) \odot P_{k+1} + \beta(\nabla f(W_{k+1}) + \varepsilon_{k+1})\|^2]$$

$$= \mathbb{E}_{\varepsilon_{k+1}}[\|(\mathbf{1} - \frac{\beta}{\tau}|\nabla f(W_{k+1}) + \varepsilon_{k+1}|) \odot P_{k+1}\|^2] + \beta^2\mathbb{E}_{\varepsilon_{k+1}}[\|\nabla f(W_{k+1}) + \varepsilon_{k+1}\|^2]$$

$$+ 2\mathbb{E}_{\varepsilon_{k+1}}\left[\left\langle(\mathbf{1} - \frac{\beta}{\tau}|\nabla f(W_{k+1}) + \varepsilon_{k+1}|) \odot P_{k+1}, \beta(\nabla f(W_{k+1}) + \varepsilon_{k+1})\right\rangle\right].$$

For the sake of simplicity, below we use $[g_{k+1}]_i := [\nabla f(W_{k+1})]_i$ to denote the $i$-th coordinate of the gradient. According to Assumption 5, the first term in the RHS of (112) can be bounded by

$$\mathbb{E}_{\varepsilon_{k+1}}[\|(\mathbf{1} - \frac{\beta}{\tau}|\nabla f(W_{k+1}) + \varepsilon_{k+1}|) \odot P_{k+1}\|^2] \tag{113}$$

$$= \mathbb{E}_{\varepsilon_{k+1}}\left[\sum_{i\in\mathcal{I}}(1 - \frac{\beta}{\tau}|[g_{k+1}]_i + [\varepsilon_{k+1}]_i|)^2[P_{k+1}]_i^2\right]$$

$$= \sum_{i\in\mathcal{I}}\left(1 - \frac{2\beta}{\tau}\mathbb{E}_{[\varepsilon_{k+1}]_i}[|[g_{k+1}]_i + [\varepsilon_{k+1}]_i|] + \frac{\beta^2}{\tau^2}\mathbb{E}_{[\varepsilon_{k+1}]_i}[([g_{k+1}]_i + [\varepsilon_{k+1}]_i)^2]\right)[P_{k+1}]_i^2$$

$$\leq \sum_{i\in\mathcal{I}}\left(1 - \frac{2\beta c\sigma}{\tau} + \frac{\beta^2(4L^2\tau^2+\sigma^2)}{\tau^2}\right)[P_{k+1}]_i^2$$

$$\leq \left(1 - \frac{3\beta c\sigma}{2\tau}\right)\|P_{k+1}\|^2,$$

where the last inequality holds because of the selection of $\beta$. To bound the third term in the RHS of (112), we use the unbiased property of $\mathbb{E}_{\varepsilon_{k+1}}[\varepsilon_{k+1}] = 0$ (Assumption 3) and $\|P_k\|_\infty \leq \tau$ (Theorem 1), which lead to

$$2\mathbb{E}_{\varepsilon_{k+1}}\left[\left\langle(\mathbf{1} - \frac{\beta}{\tau}|\nabla f(W_{k+1}) + \varepsilon_{k+1}|) \odot P_{k+1}, \beta(\nabla f(W_{k+1}) + \varepsilon_{k+1})\right\rangle\right] \tag{114}$$

$$= 2\beta\mathbb{E}_{\varepsilon_{k+1}}[\langle P_{k+1}, \nabla f(W_{k+1}) + \varepsilon_{k+1}\rangle]$$

$$- 2\mathbb{E}_{\varepsilon_{k+1}}\left[\left\langle\frac{\beta}{\tau}|\nabla f(W_{k+1}) + \varepsilon_{k+1}| \odot P_{k+1}, \beta(\nabla f(W_{k+1}) + \varepsilon_{k+1})\right\rangle\right]$$

$$= 2\beta\langle P_{k+1}, \nabla f(W_{k+1})\rangle - \frac{2\beta^2}{\tau}\mathbb{E}_{\varepsilon_{k+1}}[\langle|\nabla f(W_{k+1}) + \varepsilon_{k+1}|, (\nabla f(W_{k+1}) + \varepsilon_{k+1}) \odot P_{k+1}\rangle]$$

$$\leq 2\beta\langle P_{k+1}, \nabla f(W_{k+1})\rangle + 2\beta^2\mathbb{E}_{\varepsilon_{k+1}}[\|\nabla f(W_{k+1}) + \varepsilon_{k+1}\|^2].$$

Plugging (113) and (114) into (112) yields

$$\mathbb{E}_{\varepsilon_{k+1}}[\|P_{k+2}\|^2] \leq \left(1 - \frac{3\beta c\sigma}{2\tau}\right)\|P_{k+1}\|^2 + 3\beta^2\mathbb{E}_{\varepsilon_{k+1}}[\|\nabla f(W_{k+1}) + \varepsilon_{k+1}\|^2] \tag{115}$$

$$+ 2\beta\langle P_{k+1}, \nabla f(W_{k+1})\rangle.$$

Notice that the second term in the RHS of (115) can be upper bounded by Assumption 3 and inequality (63), and the third term can be bounded by Young's inequality

$$2\beta \langle P_{k+1}, \nabla f(W_{k+1})\rangle \leq \frac{\beta c\sigma}{\tau}\|P_{k+1}\|^2 + \frac{\beta\tau}{c\sigma}\|\nabla f(W_{k+1})\|^2. \tag{116}$$

Inequality (63), (116) and the condition $\beta \leq \frac{2}{3c\sigma}$ lead to the result

$$\mathbb{E}_{\varepsilon_{k+1}}[\|P_{k+2}\|^2] \leq (1 - \frac{\beta c\sigma}{2\tau})\|P_{k+1}\|^2 + \frac{2\beta\tau}{c\sigma}\|\nabla f(W_{k+1})\|^2 + 3\beta^2\sigma^2. \tag{117}$$

Applying the inequality

$$\|\nabla f(W_{k+1})\|^2 \leq 2\|\nabla f(W_k)\|^2 + 2\|\nabla f(W_{k+1}) - \nabla f(W_k)\|^2 \tag{118}$$
$$\leq 2\|\nabla f(W_k)\|^2 + 2L^2\|W_{k+1} - W_k\|^2$$

on (117) completes the proof. □

## H.3   Proof of Bounded $P_k$

**Lemma 7** (Bounded Saturation of $P_k$). *Suppose Assumption 1, 3 and 5 hold. If $\beta$ satisfies*

$$\beta \leq \min\left\{\frac{c\tau}{2(4L^2\tau^2 + \sigma^2)}, \frac{2}{3c\sigma}, \frac{L^2\tau^3 D}{3c\sigma^3}\right\}, \tag{119}$$

*it holds that*

$$\mathbb{E}_{\varepsilon_k}[\|P_{k+2}\|_\infty^2] \leq (1 - \frac{\beta c\sigma}{\tau})\|P_{k+1}\|_\infty^2 + \frac{41\beta L^2\tau^3 D}{c\sigma} \tag{120}$$

*and further*

$$\mathbb{E}[\|P_{k+1}\|_\infty^2] \leq P_{\max}^2 := \frac{41L^2\tau^4 D}{c^2\sigma^2}. \tag{121}$$

*Proof of Lemma 7.* Let $i^* := \arg\max_i |[P_{k+2}]_i|$. We expand the expected square norm of $P_{k+2}$ by

$$\mathbb{E}_{\varepsilon_k}[\|P_{k+2}\|_\infty^2] \tag{122}$$

$$= \mathbb{E}_{\varepsilon_k}[\|P_{k+1} + \beta(\nabla f(W_k) + \varepsilon_k) - \frac{\beta}{\tau}|\nabla f(W_k) + \varepsilon_k| \odot P_{k+1}\|_\infty^2]$$

$$= \mathbb{E}_{\varepsilon_k}[\|(\mathbf{1} - \frac{\beta}{\tau}|\nabla f(W_k) + \varepsilon_k|) \odot P_{k+1} + \beta(\nabla f(W_k) + \varepsilon_k)\|_\infty^2]$$

$$= \mathbb{E}_{\varepsilon_k}[[(\mathbf{1} - \frac{\beta}{\tau}|\nabla f(W_k) + \varepsilon_k|) \odot P_{k+1}]_{i^*}^2] + \beta^2\mathbb{E}_{\varepsilon_k}[[\nabla f(W_k) + \varepsilon_k]_{i^*}^2]$$

$$+ 2\mathbb{E}_{\varepsilon_k}\left[[(\mathbf{1} - \frac{\beta}{\tau}|\nabla f(W_k) + \varepsilon_k|) \odot P_{k+1} \odot \beta(\nabla f(W_k) + \varepsilon_k)]_{i^*}\right].$$

For the sake of simplicity, below we use $[g_k]_i := [\nabla f(W_k)]_i$ to denote the $i$-th coordinate of the gradient. According to Assumption 5, the first term in the RHS of (122) can be bounded by

$$\mathbb{E}_{\varepsilon_k}[[(\mathbf{1} - \frac{\beta}{\tau}|\nabla f(W_k) + \varepsilon_k|) \odot P_{k+1}]_{i^*}^2] \tag{123}$$

$$= \mathbb{E}_{\varepsilon_k}\left[(1 - \frac{\beta}{\tau}|[g_k]_{i^*} + [\varepsilon_k]_{i^*}|)^2[P_{k+1}]_{i^*}^2\right]$$

$$= \left(1 - \frac{2\beta}{\tau}\mathbb{E}_{[\varepsilon_k]_i}[|[g_k]_i + [\varepsilon_k]_i|] + \frac{\beta^2}{\tau^2}\mathbb{E}_{[\varepsilon_k]_i}[([g_k]_i + [\varepsilon_k]_i)^2]\right)[P_{k+1}]_{i^*}^2$$

$$\leq \left(1 - \frac{2\beta c\sigma}{\tau} + \frac{\beta^2(4L^2\tau^2 + \sigma^2)}{\tau^2}\right)[P_{k+1}]_{i^*}^2$$

$$\leq \left(1 - \frac{3\beta c\sigma}{2\tau}\right)\|P_{k+1}\|_\infty^2$$

where the last inequality holds because of the selection of $\beta$. To bound the third term in the RHS of (122), we use the unbiased property of $\mathbb{E}_{[\varepsilon_k]_{i*}}[[\varepsilon_k]_{i*}] = 0$ of Assumption 5 and $\|P_{k+1}\|_\infty \leq \tau$ in Theorem 1, which lead to

$$2\mathbb{E}_{\varepsilon_k}\left[[(\mathbf{1} - \frac{\beta}{\tau}|\nabla f(W_k) + \varepsilon_k|) \odot P_{k+1} \odot \beta(\nabla f(W_k) + \varepsilon_k)]_{i*}\right] \tag{124}$$

$$= 2\beta\mathbb{E}_{\varepsilon_k}[[P_{k+1} \odot \nabla f(W_k) + \varepsilon_k]_{i*}]$$

$$\quad - 2\mathbb{E}_{\varepsilon_k}\left[[\frac{\beta}{\tau}|\nabla f(W_k) + \varepsilon_k| \odot P_{k+1} \odot \beta(\nabla f(W_k) + \varepsilon_k)]_{i*}\right]$$

$$= 2\beta[P_{k+1} \odot \nabla f(W_k)]_{i*} - \frac{2\beta^2}{\tau}\mathbb{E}_{\varepsilon_k}[[|\nabla f(W_k) + \varepsilon_k| \odot P_{k+1} \odot (\nabla f(W_k) + \varepsilon_k)]_{i*}]$$

$$\leq 2\beta[P_{k+1} \odot \nabla f(W_k)]_{i*} + 2\beta^2\mathbb{E}_{\varepsilon_k}[[\nabla f(W_k) + \varepsilon_k]_{i*}^2].$$

Plugging (123) and (124) into (122) yields

$$\mathbb{E}_{\varepsilon_k}[\|P_{k+2}\|_\infty^2] \tag{125}$$

$$\leq \left(1 - \frac{3\beta c\sigma}{2\tau}\right)\|P_{k+1}\|_\infty^2 + 3\beta^2\mathbb{E}_{\varepsilon_k}[[\nabla f(W_k) + \varepsilon_k]_{i*}^2] + 2\beta[P_{k+1} \odot \nabla f(W_k)]_{i*}.$$

$$\leq \left(1 - \frac{3\beta c\sigma}{2\tau}\right)\|P_{k+1}\|_\infty^2 + 3\beta^2\|\nabla f(W_k)\|_\infty^2 + 2\beta[P_{k+1} \odot \nabla f(W_k)]_{i*} + \frac{3\beta^2\sigma^2}{D}$$

where the last inequality comes from Assumption 5 and variance decomposition (63). Notice that the second term in the RHS of (125) can be upper bounded by Assumption 3 and inequality (63), and the third term can be bounded by Young's inequality

$$2\beta[P_{k+1} \odot \nabla f(W_k)]_{i*} \leq \frac{\beta c\sigma}{2\tau}[P_{k+1}]_{i*}^2 + \frac{8\beta\tau}{c\sigma}[\nabla f(W_k)]_{i*}^2 \leq \frac{\beta c\sigma}{2\tau}\|P_{k+1}\|_\infty^2 + \frac{8\beta\tau}{c\sigma}\|\nabla f(W_k)\|_\infty^2. \tag{126}$$

Inequality (126) and the condition $\beta \leq \frac{2}{3c\sigma}$ lead to the result

$$\mathbb{E}_{\varepsilon_k}[\|P_{k+2}\|_\infty^2] \leq (1 - \frac{\beta c\sigma}{\tau})\|P_{k+1}\|_\infty^2 + \frac{10\beta\tau}{c\sigma}\|\nabla f(W_k)\|_\infty^2 + \frac{3\beta^2\sigma^2}{D} \tag{127}$$

$$\leq (1 - \frac{\beta c\sigma}{\tau})\|P_{k+1}\|_\infty^2 + \frac{40\beta L^2\tau^3 D}{c\sigma} + \frac{3\beta^2\sigma^2}{D}$$

$$\leq (1 - \frac{\beta c\sigma}{\tau})\|P_{k+1}\|_\infty^2 + \frac{41\beta L^4\tau^3 D}{c\sigma}$$

where the last inequality holds because the parameter is chosen as $\beta \leq \frac{L^2\tau^3 D}{3c\sigma^3}$. Telescoping over 0 to $K-1$ and using the fact that $\|\nabla f(W_k)\|_\infty^2 \leq \|\nabla f(W_k)\|^2 \leq 4L^2\tau^2 D$, we achieve that

$$\mathbb{E}[\|P_k\|_\infty^2] \leq (1 - \frac{\beta c\sigma}{\tau})^k\|P_0\|_\infty^2 + \frac{41\beta L^2\tau^3 D}{c\sigma} \leq \frac{41 L^2\tau^4 D}{c^2\sigma^2}.$$

where the last inequality holds due to the fact that $P_0 = 0$. Now we complete the proof. $\qquad\square$

### H.4 Proof of `Tiki-Taka` Descent Lemma

**Lemma 8** (Descent Lemma). *Under Assumption 1, it holds that*

$$f(W_{k+1}) \leq f(W_k) - \frac{\alpha\tau\sqrt{D}}{2\sigma}\|\nabla f(W_k)\|^2 - \left(\frac{\sigma}{2\alpha\tau\sqrt{D}} - \frac{L}{2}\right)\|W_{k+1} - W_k\|^2 \tag{128}$$

$$+ \frac{\alpha\sigma}{\tau\sqrt{D}}\|P_{k+1} - \frac{\tau\sqrt{D}}{\sigma}\nabla f(W_k)\|^2 + \frac{\alpha\sigma}{\tau\sqrt{D}}\|P_{k+1}\|^2.$$

*Proof of Lemma 8.* The $L$-smooth assumption (Assumption 1) implies that

$$f(W_{k+1}) \leq f(W_k) + \langle\nabla f(W_k), W_{k+1} - W_k\rangle + \frac{L}{2}\|W_{k+1} - W_k\|^2 \tag{129}$$

$$= f(W_k) - \frac{\alpha\tau\sqrt{D}}{2\sigma}\|\nabla f(W_k)\|^2 - \left(\frac{\sigma}{2\alpha\tau\sqrt{D}} - \frac{L}{2}\right)\|W_{k+1} - W_k\|^2$$

$$+ \frac{\sigma}{2\alpha\tau\sqrt{D}}\|W_{k+1} - W_k + \frac{\alpha\tau\sqrt{D}}{\sigma}\nabla f(W_k)\|^2$$

where the equality comes from

$$\langle\nabla f(W_k), W_{k+1} - W_k\rangle = \frac{\sigma}{\alpha\tau\sqrt{D}}\left\langle\frac{\alpha\tau\sqrt{D}}{\sigma}\nabla f(W_k), W_{k+1} - W_k\right\rangle \tag{130}$$

$$= -\frac{\alpha\tau\sqrt{D}}{2\sigma}\|\nabla f(W_k)\|^2 - \frac{\sigma}{2\alpha\tau\sqrt{D}}\|W_{k+1} - W_k\|^2$$

$$+ \frac{\sigma}{2\alpha\tau\sqrt{D}}\|W_{k+1} - W_k + \frac{\alpha\tau\sqrt{D}}{\sigma}\nabla f(W_k)\|^2.$$

From the update rule of `Tiki-Taka` and bounded saturation assumption (c.f. Assumption 4), the third term in the RHS of (129) can be bounded by

$$\frac{\sigma}{2\alpha\tau\sqrt{D}}\|W_{k+1} - W_k + \frac{\alpha\tau\sqrt{D}}{\sigma}\nabla f(W_k)\|^2 \tag{131}$$

$$= \frac{\alpha\sigma}{2\tau\sqrt{D}}\|P_{k+1} - \frac{\tau\sqrt{D}}{\sigma}\nabla f(W_k) + \frac{1}{\tau}|P_{k+1}| \odot W_k\|^2$$

$$\leq \frac{\alpha\sigma}{\tau\sqrt{D}}\|P_{k+1} - \frac{\tau\sqrt{D}}{\sigma}\nabla f(W_k)\|^2 + \frac{\alpha\sigma}{\tau^3\sqrt{D}}\||P_{k+1}| \odot W_k\|^2$$

$$\leq \frac{\alpha\sigma}{\tau\sqrt{D}}\|P_{k+1} - \frac{\tau\sqrt{D}}{\sigma}\nabla f(W_k)\|^2 + \frac{\alpha\sigma}{\tau^3\sqrt{D}}\|P_{k+1}\|^2\|W_k\|_\infty^2$$

$$\leq \frac{\alpha\sigma}{\tau\sqrt{D}}\|P_{k+1} - \frac{\tau\sqrt{D}}{\sigma}\nabla f(W_k)\|^2 + \frac{\alpha\sigma}{\tau\sqrt{D}}\|P_{k+1}\|^2.$$

Substituting (131) back into (129) yields

$$f(W_{k+1}) \leq f(W_k) - \frac{\alpha\tau\sqrt{D}}{2\sigma}\|\nabla f(W_k)\|^2 - \left(\frac{\sigma}{2\alpha\tau\sqrt{D}} - \frac{L}{2}\right)\|W_{k+1} - W_k\|^2 \tag{132}$$

$$+ \frac{\alpha\sigma}{\tau\sqrt{D}}\|P_{k+1} - \frac{\tau\sqrt{D}}{\sigma}\nabla f(W_k)\|^2 + \frac{\alpha\sigma}{\tau\sqrt{D}}\|P_{k+1}\|^2$$

which completes the proof. $\square$

## H.5 Proof of Theorem 4: Convergence of `Tiki-Taka`

With the lemmas above, we can begin the proof of Theorem 4.

*Proof of Theorem 4.* The statement that $\mathbb{E}[\|P_{k+1}\|_\infty^2] \leq P_{\max}^2 := \frac{41L^2\tau^4 D}{c^2\sigma^2}$ has been shown in Lemma 7. Therefore, it suffices to prove the convergence rate.

The proof begins by defining a Lyapunov function

$$\mathbb{V}_k := f(W_k) - f^* + \frac{\sigma}{L\tau\sqrt{D}}\|P_{k+1} - \nabla f(W_k)\|^2 + \frac{5\sigma}{Lc\tau D}\|P_{k+1}\|^2 \tag{133}$$

$$+ \frac{8}{Lc\tau\sigma}\|P_{k+1}\|_\infty^2\|\nabla f(W_k)\|^2.$$

Notice that the last term of $\mathbb{V}_k$ has the iteration

$$\mathbb{E}_{\varepsilon_{k+1}}[\|P_{k+2}\|_\infty^2\|\nabla f(W_{k+1})\|^2] \tag{134}$$

$$\overset{(a)}{\leq} \frac{1}{1-u}\mathbb{E}_{\varepsilon_{k+1}}[\|P_{k+2}\|_\infty^2\|\nabla f(W_k)\|^2]$$

$$+ \frac{1}{u}\mathbb{E}_{\varepsilon_{k+1}}[\|P_{k+2}\|_\infty^2 \|\nabla f(W_{k+1}) - \nabla f(W_k)\|^2]$$

$$\overset{(b)}{\leq} \frac{1}{1-u}\left((1 - \frac{\beta c\sigma}{\tau})\|P_{k+1}\|_\infty^2 \frac{41\beta L^2\tau^3 D}{c\sigma}\right)\|\nabla f(W_k)\|^2 + \frac{L^2\tau^2}{u}\|W_{k+1} - W_k\|^2$$

$$\overset{(c)}{\leq} (1 - \frac{\beta c\sigma}{2\tau})\|P_{k+1}\|_\infty^2 \|\nabla f(W_k)\|^2 + \frac{82\beta L^2\tau^3 D}{c\sigma}\|\nabla f(W_k)\|^2 + \frac{L^2\tau^3}{\beta c\sigma}\|W_{k+1} - W_k\|^2$$

where (a) uses $\|x + y\|^2 \leq \frac{1}{1-u}\|x\|^2 + \frac{1}{u}\|y\|^2$ for any vector $x, y \in \mathbb{R}^D$ and scalar $0 < u < 1$; (b) follows Lemma 7, bounded weight (c.f. Theorem 1), and Assumption 1; (c) specifies $u = \frac{\beta c\sigma}{2\tau}$ and the inequalities $\frac{1-\beta c\sigma/\tau}{1-u} \leq 1 - \frac{\beta c\sigma}{\tau} + u$ and $\frac{1}{1-u} \leq 2$.

By inequality (134), Lemma 5, 6 and 8, we have

$$\mathbb{E}_{\varepsilon_{k+1}}[\mathbb{V}_{k+1}] \tag{135}$$

$$= f(W_{k+1}) - f^* + \frac{\sigma}{L\tau\sqrt{D}}\mathbb{E}_{\varepsilon_{k+1}}[\|P_{k+2} - \nabla f(W_{k+1})\|^2] + \frac{5\sigma}{Lc\tau D}\mathbb{E}_{\varepsilon_{k+1}}[\|P_{k+2}\|^2]$$

$$+ \frac{8}{Lc\tau\sigma}\mathbb{E}_{\varepsilon_{k+1}}[\|P_{k+2}\|_\infty^2 \|\nabla f(W_{k+1})\|^2]$$

$$\leq f(W_k) - f^* - \frac{\alpha\tau\sqrt{D}}{2\sigma}\|\nabla f(W_k)\|^2 - \left(\frac{\sigma}{2\alpha\tau\sqrt{D}} - \frac{L}{2}\right)\|W_{k+1} - W_k\|^2$$

$$+ \frac{\alpha\sigma}{\tau\sqrt{D}}\|P_{k+1} - \frac{\tau\sqrt{D}}{\sigma}\nabla f(W_k)\|^2 + \frac{\alpha\sigma}{\tau\sqrt{D}}\|P_{k+1}\|^2$$

$$+ \frac{\sigma^2}{2L\tau^2 D}\left(\left(1 - \frac{\beta\sigma}{2\tau\sqrt{D}}\right)\|P_{k+1} - \frac{\tau\sqrt{D}}{\sigma}\nabla f(W_k)\|^2 + \frac{4L^2\tau^3 D^{3/2}}{\beta\sigma^3}\|W_{k+1} - W_k\|^2\right.$$

$$\left. + \frac{2\beta\sigma}{\tau\sqrt{D}}\|P_{k+1}\|^2 + \frac{2\beta\sqrt{D}}{\tau\sigma}\|P_{k+1}\|_\infty^2 \|\nabla f(W_k)\|^2 + 2\beta^2\sigma^2\right)$$

$$+ \frac{2\sigma^2}{Lc\tau^2 D^{3/2}}\left((1 - \frac{\beta c\sigma}{2\tau})\|P_{k+1}\|^2 + \frac{4\beta\tau}{c\sigma}\|\nabla f(W_k)\|^2 + \frac{4L^2\beta\tau}{c\sigma}\|W_{k+1} - W_k\|^2 + 3\beta^2\sigma^2\right)$$

$$+ \frac{2}{Lc\tau^2\sqrt{D}}\left((1 - \frac{\beta c\sigma}{2\tau})\|P_{k+1}\|_\infty^2 \|\nabla f(W_k)\|^2 + \frac{82\beta L^2\tau^3 D}{c\sigma}\|\nabla f(W_k)\|^2 + \frac{L^2\tau^3}{\beta c\sigma}\|W_{k+1} - W_k\|^2\right).$$

Rearranging inequality (135) above, we have

$$\mathbb{E}_{\varepsilon_{k+1}}[\mathbb{V}_{k+1}] - \mathbb{V}_k \tag{136}$$

$$\leq -\left(\frac{\alpha\tau\sqrt{D}}{2\sigma} - \frac{8\beta\sigma}{Lc^2\tau D^{3/2}} - \frac{164\beta L\tau\sqrt{D}}{c^2\sigma}\right)\|\nabla f(W_k)\|^2$$

$$- \left(\frac{\sigma}{2\alpha\tau\sqrt{D}} - \frac{L}{2} - \frac{2L\tau\sqrt{D}}{\beta\sigma} - \frac{8L\beta}{c^2\sigma\sqrt{D}} - \frac{2L\tau}{\beta c^2\sigma\sqrt{D}}\right)\|W_{k+1} - W_k\|^2$$

$$- \left(\frac{\beta\sigma^3}{4L\tau^3 D^{3/2}} - \frac{\alpha\sigma}{\tau\sqrt{D}}\right)\|P_{k+1} - \frac{\tau\sqrt{D}}{\sigma}\nabla f(W_k)\|^2 - \left(\frac{\beta\sigma^3}{L\tau^3 D^{3/2}} - \frac{\alpha\sigma}{\tau\sqrt{D}}\right)\|P_{k+1}\|^2$$

$$+ \frac{\beta^2\sigma^2}{L}\left(\frac{\sigma^2}{\tau^2 D} + \frac{16\sigma^2}{c\tau^2 D^{3/2}}\right)$$

$$\leq -\frac{\alpha\tau\sqrt{D}}{2\sigma}\left(1 - \frac{16\beta\sigma^2}{\alpha Lc^2\tau^2 D^2} - \frac{164\beta L}{\alpha c^2}\right)\|\nabla f(W_k)\|^2 + \frac{\beta^2\sigma^4}{L\tau^2 D}\left(1 + \frac{16}{c\sqrt{D}}\right)$$

where the second inequality holds because the selection of $\alpha$ and $\beta$ as follows implies the coefficients of the third to fifth term of RHS are greater than 0

$$\alpha \leq \frac{\sigma}{6L\tau\sqrt{D}}, \quad \frac{8\alpha L\tau^2 D}{\sigma^2} \leq \beta \leq \frac{c^2\sigma^4}{64\alpha L\tau^4 D}. \tag{137}$$

By taking $\beta = \frac{8\alpha L\tau^2 D}{\sigma^2}$, we bound

$$\mathbb{E}_{\varepsilon_{k+1}}[\mathbb{V}_{k+1}] - \mathbb{V}_k \tag{138}$$

$$\leq -\frac{\alpha\tau\sqrt{D}}{2\sigma}\left(1 - \frac{128}{c^2 D} - \frac{1312L^2\tau^2 D}{c^2\sigma^2}\right)\|\nabla f(W_k)\|^2 + 8\alpha^2 L\tau^2 D\left(1 + \frac{16}{c\sqrt{D}}\right)$$

$$\leq -\frac{\alpha\tau\sqrt{D}}{2\sigma}\left(1 - \frac{128}{c^2 D} - \frac{32P_{\max}^2}{\tau^2}\right)\|\nabla f(W_k)\|^2 + 8\alpha^2 L\tau^2 D\left(1 + \frac{16}{c\sqrt{D}}\right).$$

Taking expectation, averaging (138) over $k$ from 0 to $K-1$, and choosing the parameter $\alpha$ as

$$\alpha = O\left(\frac{1}{\sqrt{D(1 + 16/(c\sqrt{D}))}}\sqrt{\frac{V_0}{\sigma^2 L^2 K}}\right) \tag{139}$$

deduce that

$$\mathbb{E}\left[\frac{1}{K}\sum_{k=0}^{K-1}\|\nabla f(W_k)\|^2\right] \tag{140}$$

$$\leq 2\sigma\left(\frac{\mathbb{V}_0 - \mathbb{E}[\mathbb{V}_{k+1}]}{\alpha\tau\sqrt{D}K} + \alpha L\tau\sqrt{D}\left(1 + \frac{16}{c\sqrt{D}}\right)\right)\frac{1}{1 - 128/(c^2 D) - 32P_{\max}^2/\tau^2}$$

$$\leq 2\sigma\left(\frac{\mathbb{V}_0}{\alpha\tau\sqrt{D}K} + \alpha L\tau\sqrt{D}\left(1 + \frac{16}{c\sqrt{D}}\right)\right)\frac{1}{1 - 128/(c^2 D) - 32P_{\max}^2/\tau^2}$$

$$= O\left(\sqrt{\frac{(f(W_0) - f^*)\sigma^2 L}{K}}\frac{\sqrt{1 + 16/(c\sqrt{D})}}{1 - 128/(c^2 D) - 32P_{\max}^2/\tau^2}\right).$$

$$= O\left(\sqrt{\frac{(f(W_0) - f^*)\sigma^2 L}{K}}\frac{1}{1 - 33P_{\max}^2/\tau^2}\right)$$

where the last inequality holds when $D$ is sufficiently large. The proof is completed. $\square$

# I  Simulation Details and Additional Results

This section provides details about the experiments in Section 2.1 and 5. The analog training algorithms, including Analog SGD and Tiki-Taka, are provided by the open-source simulation toolkit AIHWKIT [27], which has Apache-2.0 license; see github.com/IBM/aihwkit. We use *Softbound* device provided by AIHWKIT to simulate the asymmetric linear device (ALD), by setting its upper and lower bound as $\tau$. The digital algorithm, including SGD, and dataset used in this paper, including MNIST and CIFAR10, are provided by PYTORCH, which has BSD license; see https://github.com/pytorch/pytorch.

**Hardware.**  We conduct our experiments on an NVIDIA RTX 3090 GPU, which has 24GB memory and maximum power 350W. The simulations take from 30 minutes to one hour, depending on the size of the model and the dataset.

**Statistical Significance.**  The simulation reported in Figure 5 is repeated three times. The randomness originates from the data shuffling, random initialization, and random noise in the analog device simulator. The mean and standard deviation are calculated using *statistics* library.

## I.1  Least squares problem

In Section 2.1 and 5.1, we considers the least squares problem on a synthetic dataset and a ground truth $W^* \in \mathbb{R}^D$, whose elements are sampled from a Gaussian distribution with mean 0 and variance $\sigma_{\text{data}}^2$. Consider a matrix $A \in \mathbb{R}^{D_{\text{out}} \times D}$ of size $D = 40$ and $D_{\text{out}} = 100$ whose elements are sampled from a Gaussian distribution with variance $\sigma_A^2$. The label $b \in \mathbb{R}^{D_{\text{out}}}$ is generated by $b = AW^*$ where $W^*$ are sampled from a standard Gaussian distribution with $\sigma_{W^*}^2$.

The problem can be formulated by

$$\min_{W \in \mathbb{R}^D} f(W) := \frac{1}{2}\|AW - b\|^2 = \frac{1}{2}\|A(W - W^*)\|^2. \tag{141}$$

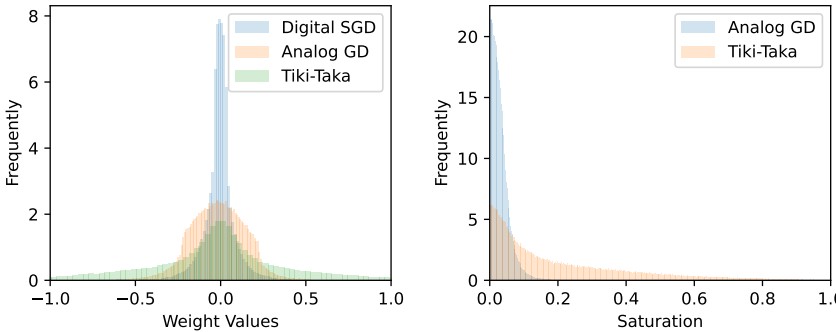

Figure 7: **(Left)** Distribution on the model weight, i.e. $\{[W_K]_i : i \in \mathcal{I}\}$. **(Right)** Saturation distribution after training, i.e. $\{|[W_K]_i|/\tau : i \in \mathcal{I}\}$.

During the training, the noise with variance level $\sigma_g^2$ is injected into the gradient. For digital SGD and the proposed `Analog` SGD dynamic, we add a Gaussian noise with mean 0 and variance $\sigma_g^2$ to the gradient. For `Analog` SGD in AIHWKIT, we add a Gaussian noise with mean 0 and variance $\sigma_g^2/s_A^2$ into $W$ when computing the gradient. The constant $s_A$ is defined as the mean of the singular value of $A$, which is introduced since the noise on $W$ is amplified by $A^\top A$, whose impact is approximately proportional to $A$'s singular value.

In Figure 1, the response step size $\Delta w_{\min}$=1e-4 while $\tau = 3$. The maximum bit length is 800. The variance are set as $\sigma_{\text{data}}^2 = 0.30^2, \sigma_A^2 = 1.00^2, \sigma_{W^*}^2 = 0.45^2, \sigma_g^2 = (0.01/D)^2$.

In Figure 3, the learning rate is set as $\alpha$ =3e-2. The response step size $\Delta w_{\min}$ is set so that the number of states is always 300, i.e. $2\tau/\Delta w_{\min} = 300$, when the $\tau$ is changing. The maximum bit length is 300. The variance are set as $\sigma_{\text{data}}^2 = 0.30^2, \sigma_A^2 = 0.50^2, \sigma_{W^*}^2 = 0.30, \sigma_g^2 = 0.10$.

## I.2 Classification problem

We conduct training simulations of image classification tasks on a series of real datasets. The gradient-based analog training algorithms, including `Analog` SGD and `Tiki-Taka`, are implemented by AIHWKIT. In the real implementation of `Tiki-Taka`, only a few columns or rows of $P_k$ are transferred per time to $W_k$ to balance the communication and computation. In our simulations, we transfer 1 column every time. The response step size is $\Delta w_{\min}$ =1e-3.

The other setting follows the standard settings of AIHWKIT, including output noise (0.5 % of the quantization bin width), quantization and clipping (output range set 20, output noise 0.1, and input and output quantization to 8 bits). Noise and bound management techniques are used in [66]. A learnable scaling factor is set after each analog layer, which is updated using digital SGD.

**3-FC / MNIST.** Following the setting in [15], we train a model with 3 fully connected layers. The hidden sizes are 256 and 128. The activation functions are Sigmoid. The learning rates are $\alpha = 0.1$ for SGD, $\alpha = 0.05, \beta = 0.01$ for `Analog` SGD or `Tiki-Taka`. The batch size is 10 for all algorithms.

**CNN / MNIST.** We train a convolution neural network, which contains 2-convolutional layers, 2-max-pooling layers, and 2-fully connected layers. The activation functions are Tanh. The learning rates are set as $\alpha = 0.1$ for digital SGD, $\alpha = 0.05, \beta = 0.01$ for `Analog` SGD or `Tiki-Taka`. The batch size is 8 for all algorithms.

**Resnet / CIFAR10.** We train different models from Resnet family, including Resnet18, 34, and 50, The base model is pre-trained on ImageNet dataset. The learning rates are set as $\alpha = 0.15$ for digital SGD, $\alpha = 0.075, \beta = 0.01$ for `Analog` SGD or `Tiki-Taka`. The batch size is 128 for all algorithms.

## I.3 Verification of bounded saturation

To further justify the Assumption 4, we visualize the weight distribution of the trained 3-FC model; see Figure 7. The results show that despite the absence of projection or saturation, the weights trained by digital SGD are bounded, which means that it is always possible to find an $\tau$ to represent all the weights in analog devices without being clipped. In the right of Figure 7, $W_{\max}$ for `Analog` SGD could be chosen as $0.2\tau$. Without constraint on the $W_{\max}$, `Tiki-Taka` has a large $W_{\max}$.

