# OpenReview forum: "Towards Exact Gradient-based Training on Analog In-memory Computing"
_NeurIPS.cc/2024/Conference — NeurIPS 2024 poster_

### Official Review · Reviewer_dVXd · 2024-06-27

**Soundness:** 3
**Presentation:** 3
**Contribution:** 3
**Rating:** 7
**Confidence:** 3

**Summary:**

This is a theoretical paper about training analog systems based on resistive memories. The paper tackles specifically the problem of weight update asymmetry in such resistive devices. The paper develops a model for the weight dynamics under “Analog SGD”, taking into account weight update asymmetry, and it shows that Analog SGD does not converge, due to an asymptotic error. The paper then shows that the Tiki-Taka algorithm, developed in prior works for such analog devices, has better convergence properties. Tiki-Taka eliminates the asymptotic error and converges to a critical point. Simulations corroborate the theory.

**Strengths:**

The topic tackled by the paper is very important. Analog computing could lead to energy efficiency gains of several orders of magnitude for both inference and training, compared to digital neural networks on GPUs. Developing appropriate theoretical models of analog computing to better understand the behavior of these analog systems is of utmost importance. To my knowledge, this paper is the first attempt to study the convergence properties of analog SGD.

The paper is clear and the study is thorough.

**Weaknesses:**

The theoretical analysis rests on a set of models and assumptions. Thus, the applicability of the conclusions rests on the validity of all these models/assumptions. In practice, I expect none of these assumptions to perfectly capture the real behavior of the system. For example, Figure 2 hints that the analog SGD dynamics (Eq3) better fits real behaviors than the digital SGD dynamics (Eq2), but the matching still seems to be far from perfect.
This, of course, is not a problem, however, what is more problematic is that there does not seem to be any discussion in the paper about which assumptions/models are accurate and widely accepted, and which ones less accurately capture empirical data or are still debated by the community.

The paper implicitly assumes that backpropagation (BP) is used for computing the weight gradients. However, a large (and growing) group of researchers in neuromorphic computing (or ”physical neural networks”) think that BP is not the best fit for analog devices, and they are exploring alternative training methods. See e.g. Ref [1] below for a very recent review of such algorithms. In particular, Ref [1] discusses several algorithms that extract or estimate the weight gradients in broad classes of analog systems. My understanding is that it wouldn’t take too much time and effort to extend the present study to include these other gradient-descent-based algorithms. In my view, including these other methods would add a lot of value to the paper, by broadening the potential impact of the work.

Reference:
[1] Momeni, Ali, et al. "Training of Physical Neural Networks." arXiv preprint arXiv:2406.03372 (2024).


Minor remark. One claim in the introduction is not properly referenced. Specifically, where do the figures of $2.4 million and $4.6 million for training LLaMA and GPT-3 come from? The references provided are the LLaMA and GPT-3 papers, which do not provide these figures, if I am not mistaken.

**Questions:**

Is the model of Eq6 for weight update asymmetry a widely accepted model? Does it match empirical data? Similarly, are the forms of q+ and q- for ALD, written at lines 174-175, widely accepted and/or corroborated by data?

Does the AIHWKit very faithfully simulate real analog devices? What are the strengths and limitations of this tool?

The paper is concerned with analog resistive memories where the weights are implemented as conductances (which are positive, I suppose), but on Figure 6 (Appendix L), I see that the weight values can be negative. Lemma 1 also indicates that w_k can be either positive or negative. Could you clarify this point?

I understand that the RHS of Eq.5 can be arbitrarily small by choosing alpha = sqrt(1/K) and letting K -> infinity. But, unless I am mistaken, this does not imply that the sequence grad f(W_k) converges to zero. Can we conclude from Eq5 that W_k converges? If not, why is Eq.5 important for studying the convergence of (digital) SGD?

I am not sure to understand why Theorem 2 is important. To my understanding, the important result is Theorem 3, which shows that analog SGD does not converge, contrary to digital SGD (Eq.5). Could you clarify this point?

Is the theory of analog SGD and Tiki-Taka provided in this paper limited to backpropagation? Or could it be used with other methods that compute weight gradients, e.g. Equilibrium Propagation and/or other algorithms presented in Ref [1] above? Similarly, could the AIHWKit be employed with these other training algorithms?

**Limitations:**

The applicability of the theoretical results depends on the validity of the underlying assumptions – see my comment above. To be clear, I am supportive of the methodology followed by the authors. However, given that the overall model will be at most as accurate as the least accurate assumption, it would be extremely useful to know which assumptions/models we can trust 100% (i.e. are well accepted), and which ones are still debated / less supported by experiments. This would help future works identify what are the main “bottlenecks” of the study, to investigate those in depth and perhaps further improve the model. For instance, can we really trust Eq3 as “the right model” of weight dynamics? Or should one merely think of it as a better model than Eq. 2?

---

> ### Author Rebuttal · Authors · 2024-08-07
>
> We really appreciate your recognizing the importance of our work and for helpful suggestions. Please find our point-by-point reply to your comments below.
> > W1. There is no discussion about which models are accurate and widely accepted.
> As the reviewer correctly pointed out, adding a discussion about the accuracy of the models are important. The applicability of our analysis relies on the following two analog hardware models:
> - Asymmetric update model (6): The update (6) omits some non-ideality, like cycle-to-cycle/device-to-device variation and analog-digital conversion error [12][27]
> - ALD response model (8): ALD models assume the response factor is a linear function, which is widely adopted in the literature [16,19,23].
>
> It explains why the proposed dynamic does not match the real behaviors perfectly in Fig 2. We will clarify the potential error raised from the simplified model in the revision. We hope the discussion helps the readers better evaluate the scope of applicability of the theory.
>
> > W2.  Including these other PNN training methods would add a lot of value, by broadening the potential impact of the work.
>
> Thank you for pointing out this interesting direction and suggesting a recent work. Indeed, this is inspiring! To establish the connection between our work and the PNN training, we will add a paragraph to briefly review the progress as follows in literature review section.
>
> > We are especially interested in AIMC hardware with resistive crossbar arrays. It is a specific implementation of physical neural network (PNN) [R1,R2]. PNN is a model implemented by a physical system involving tunable analog quantity. The quantity is adjusted to implement the learning on some specific tasks. Various hardware is capable of supporting PNN, such as holographic grating [R3], wave-based systems [R4], and photonic networks [R5], to name a few.
>
> [R1] Wright, et al. "Deep physical neural networks trained with backpropagation."
>
> [R2] Momeni, et al. "Training of Physical Neural Networks."
>
> [R3] Psaltis, et al. "Holography in artificial neural networks."
>
> [R4] Hughes, et al. "Wave physics as an analog recurrent neural network."
>
> [R5] Tait, et al. "Neuromorphic photonic networks using silicon photonic weight banks."
>
> > W3. Where do the figures of 4.6 million for training LLaMA and GPT-3 come from?
>
> Thanks for the careful reading! The cost is estimated by multiplying the required GPU hours by the AWS price per hour, where the GPU hours are reported in [1]. To avoid confusion, we will rephrase this sentence with the following one.
> > For example, it requires 184 thousand GPU hours to train an LLAMA2 7 billion model [R6], and this time increases to 1.7 million GPU hours for its 70 billion version [1].
>
> [R6] Touvron, et al. "Llama 2: Open foundation and fine-tuned chat models."
>
> > Q1. Is the model of Eq6 for weight update asymmetry a widely accepted model?
>
> The answer is affirmative. For example, [16, 52] have already demonstrated different factors scale the update in different weight state.
>
> > Q2. Does the AIHWKit faithfully simulate real analog devices?
>
> AIHWKit is capable of providing real simulations across different granular levels, including IO noise, pulse update, update variation, response factor, and A/D discretization, to name a few. One of the limitations is the significant overhand of detailed simulation and inadequate multi-GPU parallel support, which make it time-consuming to conduct large-scale simulations.
>
> > Q3. Why can the weights be negative?
> This is just for the mathematical convenience. To implement the negative weight on analog devices, two resistive crossbar arrays, a main array, and a reference array are set. The weight is represented by the difference of the two conductances multiplied by a scaling factor, which can be negative.
>
> Thanks for pointing this out! We ignored this detail in our dynamic, but we will clarify it in the revision.
>
> > Q4. Can we conclude from Eq5 that $W_k$ converges? If not, why is Eq.5 important for studying the convergence of (digital) SGD?
>
> In the convergence study of (digital) SGD, there are typically two types of convergence: the convergence to the stationary point (e.g., $W^*$ such that $\nabla f(W^*)=0$) or the convergence to the optimal solution (e.g., $W^*$ such that $W^*\in \arg\min_W f(W)$). Without additional assumptions like convexity, the convergence to the stationary point might be the best we can hope for in the worst case. Therefore, with a set of properly choosing decreasing stepsizes, Eq5 has been commonly used as a metric to assess the convergence to the stationary point of SGD; see [20]. As a result, we also use the metric of Eq5 to demonstrate the convergence of analog algorithms.
>
> > Q5. Why is Thm 2 important?
>
> Compared to Thm 3, Thm 2 is important since it serves as another main component to reveal the performance limit of Analog SGD. Thm 3 claims that there exist a *bad* situation that Analog SGD has an asymptotic error but it does not ensure it is the *worst* case. Combining Thm 2 and 3, we claim that $4\sigma^2 S_K$ is the worst-case possible asympotic error, which provide a better insight.
>
> > Q6. Is the theory provided in this paper limited to backpropagation?
>
> Thanks for the inspiring question!
> We believe our theory can be adapted to other analog training algorithms since algorithms like equilibrium propagation (EP) adopt other methods to determine the update directions. To replace the gradient in our analysis with a given update direction, we can still study the convergence using similar techniques. We will seriously consider this as a future direction.
>
> > Q7. Could the AIHWKit be employed with these other training algorithms?
>
> The answer is affirmative. AIHWKit enables multiple levels of simulation granularity. To implement other algorithms like EP, one needs to modify the algorithm-level code.
>
> With the above clarifications, we hope that our responses addressed all your questions.

---

> > ### Comment · Reviewer_dVXd · 2024-08-12
> >
> > Thank you for your detailed replies to my questions!

---

### Official Review · Reviewer_gG9b · 2024-07-11

**Soundness:** 3
**Presentation:** 3
**Contribution:** 3
**Rating:** 6
**Confidence:** 3

**Summary:**

The paper introduces the training on the analog in-memory accelerators with SGD. The traditional analog SGD algorithm suffers from inexact convergence due to asymmetric updates/gradient noise. The paper shows theoretical foundations for gradient-based training on analog devices. The authors propose Tiki-Taka, which reflects the device model to the Analog SGD. It shows better empirical performance, e.g., it can eliminate asymptotic errors and converge to critical points. They show the theoretical and empirical efficacy of the proposed approaches in overcoming training limitations on the analog devices.

**Strengths:**

- The paper is well organized and easy to follow.
- Concrete theoretical foundings on the proposed algorithm.
- Benchmark sets and the target network look reasonable, and the result also sound and look promising.
- Showing cases on various analog devices like ReRAM, etc.

**Weaknesses:**

- It'd be great if authors present how computations (ops) can be realized on the analog devices.
- Needs comparison to other analog SGD methods, like [16] or [22].

**Questions:**

- Floating-point operations are not trivial on analog devices. How can they be implemented? Additionally, for SGD training, how are analog devices more efficient compared to digital devices?
- How does it scale to large batch sizes?

---

> ### Author Rebuttal · Authors · 2024-08-07
>
> We thank the reviewer for acknowledging the merits of our work. Our point-to-point response to your comments and suggestions follow next.
>
> > W1. It'd be great if authors present how computations (ops) can be realized on the analog devices.
>
> The workload of forward and backward computation can be separated into two categories: matrix-vector multiplication (MVM) operations and non-MVM operations.
> In AIM computation hardware, the MVM operations are implemented in analog in-memory arrays, while the other non-MVM operations are conducted in digital circuits.
>
>
> We illustrate in the attached PDF in Author Rebuttal on how the MVM is implemented by analog devices. In a $3\times 3$ resistive crossbar array, a matrix $W$ is represented by the conductance of resistor at the crosspoint, where the $(i,j)$-th element of $W$ is represented by the conductance of the $(i,j)$-th resistor. To conduct an MVM operation $z = Wx$, voltage $x_{j}$ is applied between $j$-th and $(j+1)$-th row, where the subscript $j$ is used to imply the $j$-th coordinate. By Ohm's law, the current is $I_{ij}=W_{ij}x_{j}$; and by Kirchhoff's law,  the total current on the $i$-th column is
> $\sum_{j}I_{ij}=\sum_{j}W_{ij}x_{j}$.
>
>
>
> > W2. Needs comparison to other analog SGD methods, like [16] or [22].
>
> Note that the goal of this paper is not to develop the state-of-the-art analog training algorithm, but to *understand why* the vanilla SGD training on analog device does not work as expected, and why some correction operations used in the heuristic algorithms such as Tiki-Take work, through the unified lens of asymmetric updates on analog devices. We hope this theoretical understanding can contribute to future developments of analog training algorithms, hardware and material.
>
> In this context, in the introduction, we have compared the studied algorithm Tiki-Taka (TT-v1) with TT-v2 [22] and TT-v3/v4 [16] (c.f. Sec 1.2). Both [16] and [22] are variants of Tiki-Taka which were proposed to deal with some practical issues in analog training such as the reading noise and non-perfect zero-shifting issues. Since our focus in this paper is on understanding the impact of asymmetric updates on analog devices, in the simulations parts, we omitted the comparison between our paper and [16][22].
>
> Nevertheless, as requested by the reviewer, we have now compared these four methods in the same setting as that in Section 5.3. We train a CNN model under $\tau=0.7$ on MNIST dataset, whose results are listed below. TT-v2--v4 are always better than TT-v1.
>
> |Digital SGD|Analog SGD| TT-v1 | TT-v2 [22] | TT-v3 [16] | TT-v4 [16]
> |:-:|:-:|:-:|:-:|:-:|:-:|
> |99.24%|82.17%|98.56%|98.94%|98.91%|99.01%
>
> > Q1. How can floating-point (FP) be implemented by analog devices?
>
> This is a great question! The FP is implemented in the digital domain [12, 31].
> In the MVM computations, the weights are represented by conductance, while the input and output are represented by current signals. Therefore, the FP is unnecessary for MVM operations.
> The other operations involving FP computation are non-MVM ones, which are conducted in digital domains. We will add a remark in the revision to clarify it.
>
> > Q2. How does it scale to large batch sizes?
> >
>
> This is an interesting point! Implementing mini-batch gradient computation on the analog devices are significantly different from that on the digital devices. To implement the large batch in AIM computation hardware, each gradient in a batch is computed sequentially and accumulated to $W_k$ (for Analog SGD) or $P_k$ (for Tiki-Taka). For example, we use batch size 8 in FCN/CNN training and 128 for Resnet training. Studying the impact of batch size on the convergence of analog training will be an interesting future direction!
>
> We hope the above detailed clarifications can fully resolve your concerns.

---

> ### Comment · Reviewer_gG9b · 2024-08-09
>
> Thanks for the detailed response. It has clarified some of my confusion and improved my understanding on the paper. For W2, to clarify my comment, it'd be great if you can elaborate differences to existing algorithms.

---

### Official Review · Reviewer_G3ws · 2024-07-16

**Soundness:** 3
**Presentation:** 3
**Contribution:** 3
**Rating:** 5
**Confidence:** 4

**Summary:**

This paper presents a theoretical framework for gradient-based training on analog devices. This work first identifies the non-convergence problem of Analog SGD, which stems from asymptotic errors due to asymmetric updates and gradient noise and then presents a convergence analysis of Tiki-Taka, demonstrating its ability to accurately converge to a critical point, thereby eliminating the asymptotic error.

**Strengths:**

-- The proof the convergence of Analog SGD and showing that noise and asymmetric updates together lead to its asymptotic error.

-- Demonstration of Tiki-Taka algorithm precisely converging to the critical point by mitigating the drift caused by asymmetric bias and noise.

-- Empirical simulations using both synthetic and real datasets to confirm the presence of asymptotic error in Analog SGD and to show that Tiki-Taka outperforms Analog SGD.

**Weaknesses:**

-- The experimental results were obtained on rather small datasets. It'd be great to include the results on ImageNet.

-- The experiments were limited to one type of networks (i.e., resnet) on CIFAR dataset. How does it generalize on other networks such as Transformers, or mobile CNNs?

**Questions:**

See weaknesses

---

> ### Author Rebuttal · Authors · 2024-08-07
>
> We thank the reviewer for the time spent in reviewing our paper and for the valuable comments.
>
> The weaknesses identified by the reviewer mainly focus on the datasets and the model archtecture used in the experimental results  as follows.
>
> > - The experimental results were obtained on rather small datasets. It'd be great to include the results on ImageNet.
> > - The experiments were limited to one type of networks (i.e., resnet) on CIFAR dataset. How does it generalize on other networks such as Transformers, or mobile CNNs?
>
> Albeit the main focus of this paper is on the theoretical understanding of analog training, we have still conducted additional experiments during the limited rebuttal period.
>
>
> To demonstrate the efficiency of Tiki-Taka under various situations, we conducted more simulations on different datasets and model architectures, including MobileNetV2 and large/small MobileNetV3  on CIFAR10/CIRAR100 datasets.
> The results are listed as follows.
>
> Table 1: Training on CIFAR10 dataset
> ||Digital SGD|Analog SGD| Tiki-Taka
> |:-:|:-:|:-:|:-:|
> |MobileNetV2|94.47|93.88|94.24
> |MobileNetV3-Small|93.21|92.47|93.37
> |MobileNetV3-Large|94.78|94.02|94.63
>
> Table 2: Training on CIFAR100 dataset
> ||Digital SGD|Analog SGD| Tiki-Taka
> |:-:|:-:|:-:|:-:|
> |MobileNetV2|79.24|78.34|78.75
> |MobileNetV3-Small|76.41|76.13|76.45
> |MobileNetV3-Large|79.62|79.67|80.05
>
> The results show that Tiki-Taka always achieves better test accuracy than Analog SGD by about 0.5% in almost all cases, which is consistent with our conclusion. Due to the limited time and inadequate GPU resources, we can not perform simulations on a larger scale like training on ImageNet dataset during the rebuttal period.

---

### Author Rebuttal · Authors · 2024-08-07

We sincerely thank the reviewers for their constructive comments. Comments from all the reviewers were really helpful, which we believe have been fully addressed in detail in our rebuttal.

The attached PDF is the illustration to explain how analog devices implement MVM operations (see the response to Reviewer gG9b).

We look forward to the rolling dicussion and further engagement with the reviewers and area chair(s)!

---

### Decision · Program_Chairs · 2024-09-25

**Decision:**

Accept (poster)

**Comment:**

The authors analyze Tiki-Taka, an algorithm designed for analog in-memory optimization and theoretically confirm its superior performance compared to vanilla Analog SGD. The concerns raised by the reviewers have been largely addressed through authors' rebuttal and all reviewers think the paper should be accepted.